

**Characteristics of ozone and particles in the near-surface**
**atmosphere in urban area of the Yangtze River Delta, China**
Huimin Chen[1], Bingliang Zhuang[1,*], Jane Liu[1,2], TijianWang[1,**], Shu Li[1], Min Xie[1], Mengmeng
Li[1], Pulong Chen[1], Ming Zhao[1]
[1]School of Atmospheric Sciences, CMA-NJU Joint Laboratory for Climate Prediction Studies,
Jiangsu Collaborative Innovation Center for Climate Change, Nanjing University, Nanjing 210023,
China
[2]Department of Geography and Planning, University of Toronto, Toronto, M5S 3G3, Canada
Correspondence:   Bingliang   Zhuang   (blzhuang@nju.edu.cn)   and   Tijian   Wang
(tjwang@nju.edu.cn)
**Abstract**
To improve the understanding of the interactions between particles and trace gases in a typical city
of the YRD region, continuous measurements of particles and trace gases were made at an urban
site in Nanjing during cold seasons in 2016 in this study. The average of particles, including black
carbon (BC), $PM_{2.5}$, and $PM_{10}$ are $2.602 \pm 1.720$ μg/m³, $58.2 \pm 36.8$ μg/m³, and $86.3 \pm 50.8$ μg/m³,
respectively, while the average of trace gases, which contain CO, $O_3$, $NO_x$, and $NO_y$, are $850.9 \pm$
$384.1$, $37.7 \pm 33.5$, $23.5 \pm 14.7$, and $32.8 \pm 22.3$ ppb, respectively. Compared to National Ambient
Air Quality Standards in China (NAAQS-CN), we found 48 days excess of $PM_{2.5}$, 14 days excess
of $PM_{10}$, and 40 days excess of $O_3$. The particles, CO, and nitrogen oxide concentrations shared a
similar pattern of seasonality and diurnal cycles, which are different from $O_3$. The former ones are
all high in DJF and at rush hours, while the latter one had high loadings in the daytime, especially



when the ultra violet (UV) was high. Correlation analysis reveals the formation of secondary
aerosols, especially $PM_{2.5}$, under high $O_3$ and temperature conditions, and suggests a
VOC-sensitive regime for photochemical production of $O_3$ in urban Nanjing in cold seasons.
Backward trajectory analysis suggests the prevailing winds in Nanjing were northerly and easterly
during cold seasons in 2016. Air masses from eastern without passing through the urban
agglomeration and those from northern without crossing BTH regions were cleaner, but air masses
from local regions were more polluted in winter. A case study for a typical $O_3$ and $PM_{2.5}$ episode
in December 2016 demonstrated that the episode was generally associated with regional transport
and stable weather system. Air pollutants were mostly transported from the western areas with
high emissions and weather conditions are controlled by anticyclone and high-pressure system in
this region. This study further reveals the important effects of weather system and human activities
on the environment in the YRD region, especially in the urban areas, and it's an urgent need for
improving air quality in these areas.
**1. Introduction**
Particles, including black carbon (BC), $PM_{2.5}$, and $PM_{10}$, and trace gases, such as
carbon monoxide (CO), ozone ($O_3$), nitric oxide and nitrogen dioxide ($NO_x$), and total
reactive nitrogen ($NO_y$), are important components in the troposphere because of their
impacts on human health, biosphere and climate changes (e.g., Chameides et al.,
1999a, b; Jerrett et al., 2009; Allen et al., 2012). BC is mostly from incomplete
combustion of coal, diesel fuels, biofuels, and outdoor biomass burning (Bond et al.,





2004). Although BC accounts for a relatively small portion of the total mass
concentrations of aerosol particles in atmosphere, it plays a significant role in global
radiation balance, both directly and indirectly. Thus, BC could influence the global
and region climate changes and atmospheric environment substantially (Jacobson et
al., 2002; Bond et al., 2013; Deng et al., 2010). Particulate matters (PMs) originate
from both natural and anthropogenic emission sources (Kaufman et al., 2002). Due to
prosperous economic development, rapid industrialization and urbanization in recent
decades, haze events have frequently occurred in the Beijing-Tianjin-Hebei (BTH)
area, Yangtze River Delta (YRD) and Pearl River Delta (PRD) regions, all of which
were mainly caused by high concentrations of particulate matter. Tropospheric ozone
is a typical secondary air pollutant that is related to its precursors NOx and VOCs
(Crutzen, 1973) through several complicated reactions. $O_3$ could impact tropospheric
environment (Monks et al., 2015), and make significant contributions to radiative
forcing of climate (Intergovernmental Panel on Climate Change (IPCC), 2007).
Tropospheric $O_3$ precursors and the interactions between $O_3$ and its precursors in
different   geographical   locations   are   usually   different,   and   thereby   the
characterizations of $O_3$ at different sites can vary greatly (Xie et al., 2016). The impact
of PMs and BC on surface ozone is a topic that has attracted much attention. Jacobson
(1998) reported that aerosols containing BC cores reduced photolysis rates and
resulted in a decrease in ozone concentrations by 5%–8% at ground level in Los
Angeles. It is also found that a strong reduction in photolysis rate (10%–30%) due to
BC-containing aerosols (Castro et al., 2001) led to a decrease in surface ozone in



Mexico City. Similar results have been found in other studies simulating the effects of
BC on surface ozone in China (Li et al., 2011).
Most of earlier studies on particles were focused on concentrations estimation, the
chemical characteristics, potential sources, as well as climate effects of particulate
matters based on numerical simulations (Wu et al., 2012; Song et al., 2014; Xiao et al.,
2012; Yu et al., 2015; Kristjánsson, 2002; Liao and Seinfeld, 2005; Zhuang et al.,
2010; 2013), while observation-based studies of particles were relatively limited. In
addition, although a good understanding of the characteristics of $O_3$ have been gained
in the BTH area and the PRD region (Wang et al., 2009; Zheng et al., 2010; Lin et al., 2008)
due to a relatively long history of research limited in the megacities, in the YRD
region, there were only very limited studies of $O_3$ made in urban areas in some YRD
cities (Tu et al., 2007; Ding et al., 2013; Xie et al., 2016), most of which were based
on studies of $O_3$ measurement beginning in the 1990s at Lin'an site, a rural region in
the southeast YRD (Luo et al., 2000). And most of studies in YRD on particles, or
particulate matter, were done in the eastern YRD, close to Shanghai, and mainly
covered short periods of time. In the YRD region, the prevailing winds are from
between the northeast and southeast. Therefore, western YRD region is under a
downwind condition. As only few measurement studies have been conducted for
western YRD (Tu et al., 2007), large knowledge gaps still exist in our understanding
of the characteristics and main sources of $O_3$ and particles (Ding et al., 2013) in the
region, let alone their interactions.



China is always one of the major source regions of particles. Over recent decades,
along with the rapid economic development and the growing demand of energy
consumption, many areas in China are suffering from the elevated $O_3$ pollution. In the
BTH area, the YRD region, and the PRD region, all of which are the economically
vibrant and densely populated, high levels of ozone precursor emissions and $O_3$
pollution have become one of the major environment problems affecting the public
(Chan and Yao, 2008; Zhang et al., 2009; Ma et al., 2012; Xie et al., 2016). Because
of complex sources and chemical reactions, and relatively long atmospheric lifetimes
of the pollutants in the atmosphere that favors regional and long-range transport, all
the pollutants are of great concern for regional air quality but are very difficult to
control (Cooper et al., 2005; Zhang et al., 2008). The YRD is located in the eastern
part of the Yangtze River Plain, adjacent to the most polluted North China Plain,
including large cities of Shanghai, southern Jiangsu and northern Zhejiang. Taking up
only 2 percent of the land area in China, this region produces over 20 percent of
China's Gross Domestic Product (GDP). Nanjing, as the capital of Jiangsu Province,
lies in the middle of YRD. It covers an area over 6000 $km^2$, with more than 7.3
million residents (http://www.njtj.gov.cn/). Being the second largest commercial
center after Shanghai in YRD, even the East China, Nanjing is highly urbanized and
industrialized. Both particles and $O_3$ concentrations are found to be high in Nanjing,
which affects regional climate and air quality (Zhang et al., 2009; Yi et al., 2015).
Therefore, the issue of air pollution in Nanjing deserves attentions. In this study,
continuous observations of particles, trace gases and certain aerosol optical properties



at an urban station in Nanjing (a typical developing city in YRD) have been made in
order to characterize the air pollution in the city. In the following, we describe the
methodology in Section 2. Results and discussions are presented in Section 3,
followed by Conclusions in Section 4.
**2. Methodology**
**2.1 Brief Introduction to the Urban Atmospheric Observational Station**
The Urban Atmospheric Observational Station is a regional atmospheric urban station
located on the Gulou Campus of Nanjing University in the downtown area of Nanjing
(32.05 °N,118.78 °E), and run by School of Atmospheric Sciences, Nanjing
University. It is built on the roof of a 79.3m tall building, without any industrial
pollution sources within a 30 km radius around but several main roads with evident
traffic pollution, especially during rush hours. The sketch map of the site (not shown)
and the corresponding climatology have been described in Zhu et al (2012).

The Particles, $O_3$, $NO_x$, $NO_y$, CO, and wavelength-dependent aerosol optical
parameters including aerosol scattering (SC), back-scattering (Bsp), and absorption
(AAC) coefficients have been routinely measured at the station during the time period
from 1 Sep 2016 to 28 Feb 2017. The AAC and concentrations of BC were derived
from the measurements using a seven-channel Aethalometer (model AE-31, Magee
Scientific, USA). The aerosol SC and Bsp were measured with a



three-wavelength-integrating Nephelometer (Aurora 3000, Australia). The AE-31
model measures light attenuation at seven wavelengths, including 370, 470, 520, 590,
660, 880 and 950 nm, with a desired flow rate of 5.0 L min$^{-1}$ and a sampling interval
of 5 min. Aurora 3000 measures aerosol light scattering, including SC and Bsp at 450,
525 and 635 nm, with a sampling interval of 1 min (Zhuang et al. 2017). Precision
and instrument of the measurements in this study are listed in Table 1.

Monthly averaged meteorological parameters during the period from Sep.2016 to
Feb.2017 at the station are shown in Table 2. The air temperature at the site ranged
from 6.64℃ in Feb.2017 to 24.88℃ in Sep.2016. The relative humidity (RH) was
higher in fall than in winter, especially in October, while the precipitation was heavier
in fall than in winter. Visibility (Vis) varied in different months. The peak of the
ultraviolet radiation (UV) occurred in Sep.2016, after which the radiation became
weak till the end of Jan.2017, and rose a little in Feb.2017.

**2.2 Calculation of the aerosol optical properties**
The wavelength-dependent AAC, which is associated with the intensities of the
incoming light and remaining light after passing through a medium, can be calculated
directly using the measured light attenuations (ATN) through a quartz filter matrix, a
percentage to represent the filter attenuation, as well as BC mass concentrations
(Petzold et al., 1997; Weingartner et al., 2003; Arnott et al., 2005; Schmid et al.,



150  2006):

$$\sigma_{ATN,\,t(\lambda)} = \frac{(ATN_t(\lambda)-ATN_{t\text{-}1}(\lambda))}{\Delta t} \times \frac{A}{V} \quad,$$ (1)
where $A$ (in m$^2$) is the area of the aerosol-laden filter spot, $V$ is the volumetric
sampling flow rate (in L min$^{-1}$) and $\Delta t$ is the time interval (=5 min) between $t$ and
$t$-1. $\sigma_{ATN}$, known as AAC without any correction, is larger than the actual aerosol
absorption coefficient $\sigma_{abs}$ in general. The key factors leading to the bias are as
follows: (1). multiple-scattering of light at the filter fibers (multiple-scattering effect),
and (2) the instrumental response with increased particle loading on the filter
(shadowing effect). The former results in the overestimation of the $\sigma$, while the later
causes underestimation of the $\sigma$. Thus, the correction is needed and the calibration
factors C and R (shown in Eq. 2) are introduced against the scattering effect and
shadowing effect, respectively:
$$\sigma_{abs,\,t(\lambda)} = \frac{\sigma_{ATN,\,t(\lambda)}}{C \times R} \quad.$$ (2)
Weingartner (Weingartner et al., 2003, WC2003 for short, hereinafter), Arnott (Arnott
et al., 2005), Schmid (Schmid et al., 2006, SC2006 for short, hereinafte), and Virkkula
(Virkkula et al., 2007) corrections, have been developed to eliminate the uncertainties.
Zhuang et al. (2015) further suggested that wavelength-dependent AACs corrected by
SC2006 might be closer to the real ones than WC2003s in Nanjing, although 532 nm
AACs from these two corrections are close to each other.
Therefore, AACs corrected from SC2006 are used in this study.




Measurement of Aurora 3000, a nephelometer with newly designed light sources
based on light emitting diodes, needs correction using Mie-theory for measurement
artefacts. Müller et al. (2011) provided parameterizations for the angular sensitivity
functions of Aurora 3000, which follows the definition of correction factors from
Anderson and Ogren (1998), where the ratios of true to measured nephelometer
values for both total scattering and backscattering are defined by:
$$C_{ts,\lambda} = \frac{\sigma_{ts,\lambda}^{true}}{\sigma_{ts,\lambda}^{neph}} \frac{\sigma_{tsR,\lambda}^{neph}}{\sigma_{tsR,\lambda}^{true}} \quad , \tag{4}$$
and
$$C_{ts,\lambda} = \frac{\sigma_{bs,\lambda}^{true}}{\sigma_{bs,\lambda}^{neph}} \frac{\sigma_{bsR,\lambda}^{neph}}{\sigma_{bsR,\lambda}^{true}} \quad , \tag{5}$$
where $\sigma_{ts}^{true}$ and $\sigma_{bs}^{true}$ are true total scattering coefficient and backscattering
coefficient for ideal angular sensitivity functions, respectively, $\sigma_{tsR,\lambda}$ and $\sigma_{bsR,\lambda}$ are
Rayleigh total scattering coefficient and backscattering coefficient, respectively, and
$\sigma_{ts}^{neph}$ and $\sigma_{bs}^{neph}$ are nephelometer total scattering coefficient and backscattering
coefficient, respectively. In this study, we assume that Rayleigh scattering is
equivalent to true scattering.

The correction factors can be calculated using measured size distributions or SAE.
Anderson and Ogren (1998), hereinafter denoted as AO98, found a dependency
between the SAE and the correction factor for total scattering. The correction was
given by:
$$C_{ts} = a + b \cdot \alpha_{ts}^{*} \tag{6}$$




where $\alpha_{ts}^{*}$ is the scattering Ångström exponent derived from uncorrected nephelometer
scattering. According to Müller et al. (2011), for backscattering, there was no
correlation between correction factors and scattering Ångström exponents, which is in
agreement with AO98. The parameters $a$ and $b$ were derived from Mie calculated
true scattering and simulated nephelometer scattering for ranges of particle sizes and
refractive indices.

In this study, we used the correction factors for Aurora 3000 without a sub-μm cut in
Müller et al. (2011), which are shown in the Table 3. According to nephelometer
correction factors for angular nonidealities, which are shown in Table 3(a), original
scattering coefficient (SC at 635 nm, 525 nm and 450 nm ) and backscattering
coefficient (Bsp at 635 nm, 525 nm and 450 nm) obtained from the measurements are
corrected based on Eqs (4) and Eqs (5). We also calculated correction factors for total
scatter as function of Ångström exponent shown in Table 3.(b), original scattering
coefficient (SC at 635 nm, 525 nm and 450 nm ) are corrected based on Eqs (6).

Based on corrected wavelength-dependent AAC and SC, SAE and AAE are estimated
by the following:
$AAE_{470/660nm} = -\log(AAC_{470nm}/AAC_{660nm})/\log(470/660)$,  (7)
$SAE_{450/635nm} = -\log(SC_{450nm}/SC_{635nm})/\log(450/635)$,  (8)
$\sigma_{\lambda} = \sigma_{\lambda_0} \times (\frac{\lambda}{\lambda_0})^{-\alpha}$ ,  (9)



where $\sigma_\lambda$ is the coefficient at wavelength $\lambda$ and $\alpha$ is the corresponding Ångström
exponents.

On the basis of Eqs (7) ~Eqs (9), SC and Bsp at 550 nm were calculated for
comparison. Between the two ways of corrections, the results of the total scattering
coefficients are in agreement with each other in general, with differences of 10.67%.
In this study, we choose the results from the correction using SAE.
Meanwhile, based on wavelength-dependent SC, Bsp, AAC, aerosol asymmetry
parameter (ASP), single-scattering albedo (SSA) and extinction coefficient (EC) are
further estimated:
$$ASP_\lambda = -7.143889\beta_\lambda^3 + 7.46443\beta_\lambda^2 - 3.9356\beta_\lambda + 0.9893, \qquad (10)$$
$$SSA_\lambda = \frac{SC_\lambda}{SC_\lambda + AAC_\lambda}, \qquad (11)$$
$$EC_\lambda = SC_\lambda + AAC_\lambda, \qquad (12)$$
where is $\beta_\lambda$ the ratio of Bsp to SC at wavelength $\lambda$. Equation (10) is derived from
Andrews et al. (2006).
Table 4 shows the statistical summary of the surface aerosol optical properties in
Nanjing after the correction. The mean value during the cold seasons in 2016 of AAC,
SC, Bsp, EC, SSA and ASP at 550 nm, AAE at 470/660 nm and SAE at 450/635 nm
are 23.741, 349.502, 35.469, 373.536 Mm$^{-1}$, 0.929, 0.645, 1.600, and 1.192, with a
standard deviation of 15.556, 235.291, 21.488, 247.877 Mm$^{-1}$, 0.028, 0.052, 0.175,
and 0.288, respectively.



## 2.3 HYSPLIT model

In order to understand the general transport characteristics of air masses recorded at

this station, we conducted a 4 d (96 h) backward trajectory simulations during the

cold seasons in 2016 using a Lagrangian dispersion model Hybrid Single-Particle

Lagrangian Integrated Trajectory (HYSPLIT) (version 4.9) provided by the Air

Resource Laboratory (ARL) of the USA National Oceanic and Atmospheric

Administration (NOAA) (Draxler and Hess, 1998). HYSPLIT - 4 Model is capable of

processing multiple gas input fields, multiple physical processes and different types of

pollutant emission sources and has been widely used in the study of transport and

diffusion of various pollutants in various regions (Mcgowan and Clark, 2008; Wang

et al., 2011; Wang et al., 2015). It is one of the most extensively used atmospheric

transport and dispersion models for the study of air parcel trajectories (Draxler and

Rolph, 2013; Stein et al., 2016). In this study, backward trajectories were calculated

and clustered using a stand-alone version of the NCEP / NCAR reanalyzed

meteorological field (http://ready.arl.noaa.gov/archives.php). The NCEP data contain

6-hourly basic meteorological fields on pressure surfaces, with the spatial resolution

of 2.5°, corresponding to the 00, 06, 12, 18 UTC, respectively. In this study, the data

are also converted to hemispheric 144 by 73 polar stereographic grids, which is the

same grid configuration as the dataset applied in synoptic weather classification. For

each synoptic weather pattern, the backward trajectories were started at Gulou station

in Nanjing (32°N, 118.8°E).



## 3. Results and discussion

### 3.1 Characteristics of particulate matter in Nanjing

The hourly-mean concentrations of particles at Gulou site during the cold seasons in
2016 are shown in Fig 1. Gaps in the time series are missing values. The averaged
values of BC, $PM_{2.5}$ and $PM_{10}$ during the study period are $2.6 \pm 1.7$, $58.2 \pm 36.8$, and
$86.3 \pm 50.8$ μg/m³, respectively. The average of particulate matter concentrations
during the study period are higher than standard concentrations, which are 35 μg/m$^3$
for fine and 70 μg/m$^3$ for $PM_{10}$. Particles, including BC, $PM_{2.5}$ and $PM_{10}$ fluctuate
similarly, because the three particules originate mostly from the same sources, i.e.,
transport emissions. BC loadings at Gulou were low in September and October,
usually below 6 μg/m³, while the loadings were high in the other months, such as in
mid-November, early and late December, early January, and mid-to-late February,
suggesting occurrences of BC pollution events during these periods. $PM_{2.5}$ loadings
and $PM_{10}$ loadings were generally below 120 and 200 μg/m³, respectively, but higher
during early October and in the periods when BC loadings were high. The particle
concentrations are affected by various factors and progress. For example, the high
loadings of particulate matter in early October was mainly due to the increase in
aerosols concentrations with high scatter coefficient (SC), and thus the BC loadings
did not show such peak during early October.

Monthly variations of particles in the cold seasons in 2016 were obvious (Fig.2). High





particle concentrations were observed from November to February while the low ones
were in September and October. The smallest monthly concentrations of BC, $PM_{2.5}$,
and $PM_{10}$ occurred in October, being 1.8, 39.2, and 59.8 μg/m³, respectively, while
the largest monthly concentrations occurred in December, being 3.7, 85.0, and 123.1
μg/m³, respectively, which were about twice of those in October. In general, there are
two key factors that impact particle concentrations: meteorology and emissions.
Heavy precipitation with a strong scavenging effect in October might directly lead to
small loadings of particles (Table.2). Anthropogenic particle emissions from fossil
fuel over China increased after summer and showed a sharp increase from November
to January (Zhang et al., 2009), which may explain the high particle concentrations
during those periods. Qian et.al (2014) believed that high particle loadings in Nanjing
from late October to early November resulted from the large-scale burning of crop
residues. However, $PM_{2.5}$ and $PM_{10}$ concentrations reached a relative maximal in
early October, while the emission in October is relative low compared to the
following months (Zhang et al., 2009).

Substantial diurnal cycles of the particles are also observed (Fig.3). BC levels were
high at rush hours (7~9 am and 8~11 pm) but low in afternoon (1~3 pm). Zhuang et al.
(2014) mentioned that high BC concentrations in these times of the day might be
caused by the vehicle emissions (as mentioned in Section 2, several main roads with
apparent traffic pollution surround the station). In addition, temperature was low after
midnight, and the atmosphere stratification was stable. Therefore, it was easy for



temperature inversion to appear, which was not conductive to the diffusion of
pollutants, and the concentrations of particles accumulated and reached a peak at
around 8 am. Atmosphere stratification became stable again as the temperature
decreased after around 4 pm, which may also explain the peak during 9~11 pm (Qian
et al., 2014). As to the low BC in afternoon, which occurred at around 3 pm, it was
mainly induced by well-developed boundary layer. Because the atmosphere became
less stable with increasing temperature, and strong turbulent exchange and vertical
diffusion were favorable to the diffusion of pollutants, BC concentrations decreased to
a minimum in the afternoon. Fig. 3 also shows that the peak values of fine particle
concentrations often occurred one or two hours later than those of BC concentrations,
with high values at around 10 am and low values at around 5 pm. According to
Khoder (2002), atmospheric photochemical reactions are extremely active under
conditions of strong radiation and high temperature, during which more secondary
aerosol particles (like sulfate particles) generated, so the concentrations of fine
particles in the atmosphere will increase. When solar radiation was strong, ultra-fine
particles generated during photochemical reactions contributed greatly to the
concentrations of aerosol particles.
Generally, the diurnal cycles of BC had a bimodal distribution with two peaks, while
$PM_{2.5}$ and $PM_{10}$ had only one peak. However, both magnitude and temporal variations
of particles were changed in winter, and there is another peak at around 2 am(see S1),
which was possibly due to the affection of BC pollution episodes at night.



**3.2 Characteristics of gaseous pollutants in Nanjing**


Fig.4 shows hourly-mean concentrations of gaseous pollutants at Gulou during the
cold seasons in 2016, in which, there were few gaps for invalid values. The averaged
concentrations during the fall and winter of CO, $O_3$, $NO_x$ and $NO_y$ at the site are 851
$\pm$ 384, 37.7$\pm$ 33.5, 23.5$\pm$ 14.7, and 32.8$\pm$ 22.3 ppb, respectively. As shown in Fig.4,
$O_3$ concentrations in the site were extremely high during the entire September in 2016,
with a maximum over 200 ppb, which was mainly due to the strong solar radiation
and the high temperature lasting in September. $O_3$ concentrations began to increase in
February because of enhanced solar radiation, after a low-concentration period since
late October, during which $O_3$ concentrations were below 100 ppb. $NO_x$ and $NO_y$
have a similar pattern: the concentrations were high in November, December and
February (Fig.5). It is noticeable that the daily variation of CO concentrations was
similar to that of BC. A remarkable correlation between BC and CO is found in a
number of studies (Jennings et al., 1996; Derwent et al., 2001; Badarinath et al., 2007;
Spackman et al., 2008), suggesting that both of the pollutants are greatly affected by
anthropogenic sources and biomass burning in eastern China.

Fig.5 illustrates monthly variations of $O_3$, nitrogen oxides ($NO_y$ and $NO_x$), and CO in
the cold seasons in 2016. $O_3$ peaked in September at 74.8 ppb while $NO_y$ and $NO_x$
peaked in December at 31.8 and 41.7 ppb, respectively. $O_3$ reached minimum at 23.4
ppb in November and $NO_y$ and $NO_x$ ware lowest in September, being 14.5, and 20.8





ppb, respectively. $O_3$ is a secondary pollutant and complicatedly related to its
precursors, including $NO_x$ and VOCs. $O_3$ precursors and their effects on $O_3$ formation
are different at different geographical locations, and thus the characterizations of $O_3$ at
different sites can vary greatly. $O_3$–$NO_x$–VOCs relationships can be described by the
following reactions:
$O(^3P) + O_2 + M \rightarrow O_3 + M$ \hfill (R1)
$NO_2 + h\upsilon \rightarrow NO + O(^3P)$ \hfill (R2)
$O_3 + NO \rightarrow O_2 + NO_2$ \hfill (R3)
$HO_2 + NO \rightarrow OH + NO_2$ \hfill (R4)
$RO_2 + NO \rightarrow OH + NO_2$ \hfill (R5)
$OH + RH + O_2 \rightarrow RO_2 + H_2O$ \hfill (R6)
$RO + O_2 \rightarrow HO_2 + carbonyls$ \hfill (R7)
where (R4), (R5), and (R2) reactions establish an "$NO_X$ cycle" that could produce $O_3$
without consumption of $NO_X$, the other important chemistry cycle is the so-called
"$RO_X$ ($RO_X$=OH+$HO_2$ + $RO_2$) radical cycle" that could continuously supply $HO_2$ and
$RO_2$ to oxidize NO to $NO_2$, and (R7) is usually referred as $NO_x$ titration, which is an
important $O_3$ removal process related to freshly emitted NO. In general, when $NO_x$
concentrations were high, $O_3$ concentrations may experience a depression process
since excessive NO are not favorable for the $O_3$ production (Xie et al, 2016; Wang et
al., 2018). The CO concentrations varied greatly in winter because of the frequent
shifting of air masses from the clean interior continent and heavily polluted urban
plumes in the heating period (normally from November to March in Northern China,

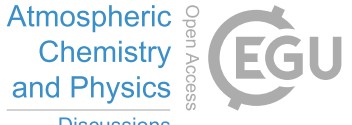

(Pan et al., 2011). In September and October, the CO concentrations at Gulou
apparently decreased owing to frequent intrusions of clean air mass from the Pacific
Ocean, and this seasonal trend was confirmed by HYSPLIT-4 model (see detailed
discussion in Section.4).

Fig. 6 (a) shows the mean diurnal variations of the gaseous pollutants ($O_3$, $NO_x$, $NO_y$,
and CO) at Gulou during the cold seasons in 2016. The concentrations of $O_3$ were the
lowest around 7 am and went up rapidly corresponding with the increase of solar
radiation. After reaching the peak in the middle of the day at 3 pm, the $O_3$
concentrations kept decreasing rapidly until sunset. During the nighttime, the
concentrations of $O_3$ decreased slowly and maintained low values, attributed to the
process of $NO_x$ titration and the lack of solar radiation. With respect to $NO_x$ and $NO_y$,
two peaks appeared in the diurnal cycles, one around 9 am and the other at 8 pm. Both
peaks coincided with the rush hours in the city, during which large amounts of vehicle
emissions were released. The morning peak was slightly higher than the evening one.
The abovementioned diurnal cycles in $O_3$ and nitrogen oxides concentrations followed
the typical patterns in polluted areas (Lal et al., 2000; Lam et al., 2001; Wang et al.,
2006; Tu et al., 2007; Ding et al., 2013; Xie et al, 2016). Since CO concentrations
showed a diurnal variation similar to that of BC, the two peaks around 9 am and 8~10
pm could also result from vehicle emissions at rush hours.

According to Xie et al. (2016), these diurnal variation patterns of $O_3$ and $NO_x$ are





mainly resulted from the photochemical processes and the meteorological conditions.
The ultraviolet irradiance (UV) at Gulou started to increase at about 7 am (Fig.6 (b)),
which could induce a series of photochemical reactions including the formation of
peroxy radicals ($HO_2$ and $RO_2$ etc.) and the photolysis of $NO_2$. From 8 am to 3 pm,
the increase in UV enhanced the $O_3$ formation by the production processes of (R4)–
(R5). Simultaneous measurement of $O_3$ and UV shows that the $O_3$ concentrations are
highly correlated to UV, with a correlation coefficient of 0.47. It is also noticeable
that the $O_3$ maximum was 2 h after the UV maximum, suggesting the time to take for
the chemical reactions. The slightly reduction of $O_3$ and $NO_x$ after the midnight is
likely due to of $NO_x$ titration. The development of the planetary boundary layer (PBL)
can also modulate pollutant concentrations. The concentrations of a pollutant are
diluted when PBL rise during the daytime and enhanced in the low nocturnal PBL that
favors pollutant accumulation, after comparing Fig.6 (a) with the reported diurnal
variation of PBL height in Nanjing (Jiang et al., 2014; Xie et al., 2016).

Table 7 further provides the statistics of $O_3$, $PM_{2.5}$ and $PM_{10}$ mass concentrations with
a comparison to the National Ambient Air Quality Standards in China (NAAQS-CN),
which were released in 2012 by the China State Council and will be implemented
nationwide in 2016 (MEP, 2012). According to NAAQS-CN for $PM_{2.5}$ and $PM_{10}$ (75
μg/m$^3$ of $PM_{2.5}$ concentrations and 150 μg/m$^3$ of $PM_{10}$ concentrations for 24h average),
there were 48 days of $PM_{2.5}$ exceedances, accounting for about 30% of the 6 months
period, and 14 days of $PM_{10}$ exceedances, lower than the $PM_{2.5}$ exceedances. Days of





particulate matter exceedances mainly occurred during DJF. Donkelaar et al. (2010)
reported that a multi-year average of $PM_{2.5}$ mass concentrations was over 80 μg/m$^3$ in
eastern China by using satellite data during 2001–2006, and Ding et al. (2013) stated
an 1-year average about 75 μg/m$^3$ in rural area of Nanjing form August 2011 to July
2012. Therefore, the means in Table 7 show lower particle concentrations than what
were reported. The days of exceedances also were fewer than in 2011 (Ding et al.,
2013), during which 99 days of $PM_{2.5}$ exceedances happened during the cold seasons.
These results suggest that particles control policies are well-implemented in Nanjing
although particles remain a severe pollution problem in the YRD region. According to
NAAQS-CN for $O_3$ (160 μg/m$^3$ for 8 h average and 200 μg/m$^3$ for 1 h average), 37
days of exceedances occurred (Table7), covering 20% of the period and mostly
appearing in September and February when the air temperature was relatively high. In
contrast to particulate matter, $O_3$ concentrations increased from 2011 to 2016, and the
exceedance days were 10 times of those in 2011. It was found in previous studies that
$O_3$ levels in the rural areas were generally higher than those in the city centers (Zhang
et al., 2008; Geng et al., 2008; Xie et al., 2016). Thus, high $O_3$ concentrations and
severe air pollution in Gulou, an urban site, suggest a severer $O_3$ pollution problem in
the entire YRD region. Note that this study only discussed the $O_3$ concentrations in
the cold seasons when the concentrations of $O_3$ are lower than in the warm season,
suggest the problem can be severer in the warm seasons. The emissions of $O_3$
precursors (VOCs and $NO_y$) in Nanjing have significantly increased with the
increases of residents (over 200,000 increase since 2011), the number of automobiles



(over 65% increase since 2011), and GDP (gross domestic production) (nearly 70%
increase since 2011). Consequently, $O_3$ concentrations at ground level has gradually
risen (http://www.njtj.gov.cn/).

**3.3 Inter-species correlations**
Correlations between different species were analyzed to help interpret the data and
gain insights into the underlying mechanisms/processes. Because precipitation could
impact wet scavenging processes for particles and other aerosols (see S2), we
eliminated the data in rainy condition.

The scatter plot of $O_3$ measured at the site and $NO_x$ color-coded with air temperature
is given in Fig.7 (a). The negative correlation suggests a titration effect of freshly
emitted NO with $O_3$ in the cold seasons. In addition, the slope decreased when air
temperature rose. These results suggest a strong photochemical production of $O_3$ in
this region during high air temperature, resulting in the seasonal cycle pattern of $O_3$
shown in Fig. 5 (a) (Ding et.al, 2013). Previous research has shown that visibility has
a good correlation with the concentrations of particles and relative humidity. With an
increase in the $PM_{2.5}$ concentrations, visibility decreases exponentially (Fig.7 (b)),
because the concentrations of particles would increase scattered and absorption
extinction coefficients, while the visibility (Vis) is related to the coefficients through:
$$Vis = \frac{3.91}{\sigma} \tag{13}$$





where $Vis$ is the visibility and $\sigma$ is the extinction coefficient (EC) (Larson et.al,
1989). As for the effect of relative humidity (RH) on the visibility, according to Mie
theory, with the increase of the relative humidity, the radius of the wet particle
increase, and so the extinction coefficient increases. Therefore, the visibility decrease.
Moreover, when RH≤ 80%, the effect of particle concentrations on visibility could
become smog, and when 80% < RH ≤ 90%, the effect could form smog and fog at the
same time. Thus, we performed a linear fit of the visibility with differing
concentrations of $PM_{2.5}$ when RH ≤ 70%, 70% <RH ≤ 80%, and 80% < RH ≤ 90%, to
find out the relationship among these factors. Although there is no precipitation, there
are still water droplets in the air when RH >90%, which become fog. Therefore, we
eliminated those data. It is found that the fitting curves are as follows: $[PM_{2.5}]$ =
$366.72[Vis]^{-0.745}$ ($R^2$ = 0.7196), $[PM_{2.5}] = 337.16[Vis]^{-0.855}$ ($R^2$ = 0.8692), and $[PM_{2.5}]$
$= 248.6[Vis]^{-0.852}$ ($R^2$ = 0.8279).

To figure out the interaction between particles and $O_3$, we give scatter plots of $PM_{2.5}$–
$O_3$ and BC–$O_3$ (Fig.8), in which data points are color-coded with air temperature.
Overall, particulate matters and black carbon are negatively correlated with $O_3$,
because particulates inhibit the photolysis reactions near the surface, reducing the
photolysis frequencies in the atmosphere, and resulting in the decrease of $O_3$
concentrations near the ground, which is also addressed using the chemical transport
model (HANK) (Li et al., 2005). It is noticeable that a negative correlation could be
found for low air temperature samples while a pronounced positive correlation existed





for high temperature data points. The negative correlation for cold air is mainly due to
the titration effect of high NO concentrations, which was associated with high
primary $PM_{2.5}$ in the cold seasons as well. And the positive correlation for high air
temperature is related to the formation of secondary fine particles associated with high
concentrations of $O_3$, which may be related to high conversion rate of $SO_2$ to sulfate
under high concentrations of oxidants (Khoder, 2002). Previous studies of $PM_{2.5}$
chemical compositions in Shanghai and Nanjing (Wang et al., 2002, 2006) suggested
that sulfate was the most dominate ion in $PM_{2.5}$. The detailed mechanisms still need to
be further addressed by long-term measurement of aerosol chemical composition.
Since black carbon is insoluble in polar and non-polar solvents and remains stable
when air or oxygen is heated to $350 \sim 400°C$, it cannot be generated nor cleared
through chemical reactions. Thus, when air temperature rises, the correlation between
BC and $O_3$ becomes obscurer compared to the one between $PM_{2.5}$ and $O_3$. Scatter
plots of $CO–NO_x$, $PM_{2.5}–NO_x$, and $BC–NO_x$, are given in Figs. 9(a)-9(c), with data
points color-coded with concentrations of $O_3$. Fig.9 (b) and 9(c) show a good positive
correlation between $PM_{2.5}$ and $NO_x$, as well as BC and $NO_x$, suggesting that the
particles at the site in Nanjing University Gulou Campus were mainly associated with
combustion sources, which is also the reason for the negative correlation between
particles and $O_3$. It is found that high $O_3$ levels are generally associated with air
masses of high $CO/NO_x$ or particles/$NO_x$ ratio, and when $NO_x$ concentrations was
lower than 40 ppb, an increase in CO or particular matter concentrations would cause
a sharp increase in $O_3$ concentrations while $NO_x$ reverses. As discussed in Atkinson





et.al (2000), volatile organic compounds (VOCs) generally have good correlation with
CO and play a role similar to CO in the photochemical ozone production. Particles
also have good correlation with CO, so the particles–$O_3$–$NO_x$ relationship may
indicate a VOC-sensitive regime of $O_3$ formation in this region, as the CO–$O_3$–$NO_x$
relationship. Geng et al. (2008) reported a VOC-sensitive regime in Shanghai by
using measured and modeling results, and Ding et al. (2013) also reported a
VOC-sensitive regime in rural area in Nanjing.

Correlations of $PM_{2.5}$–$O_3$ in daytime when UV radiation is relatively strong and
nighttime when UV radiation is approximately 0 are shown in Fig.10. It is found that
the correlation is better with a clearer tendency and $O_3$ are higher during daytime,
suggesting strong photochemistry progresses during daytime. Some data in the
nighttime plots show relatively high $O_3$. Most occurred in September and February
when $O_3$ concentrations were extremely high. It is also found that some show
relatively high $NO_x$ associated with relatively low $PM_{2.5}$. After a further backward
trajectories analysis (Section 3.4), we found that these data are most likely
corresponded to air masses coming from the nearby and northwest in November and
December, which may contain high $NO_x$ plumes and transport to Nanjing during
nighttime.
**3.4 Backward Trajectories Analysis**
The cluster means of the backward trajectories at 100 m from Gulou, Nanjing, in 2016



fall (Fig.11) and winter (Fig. 13) suggest different air flows that were transported to
Nanjing from long distances. Most of air masses came from the oceans in fall (40 %,
cluster 4 in Fig. 11) and from the north and north-west of China in winter (49 %,
clusters 1 and 4 in Fig. 13). Although air masses came from north in both fall (cluster
4) and winter (cluster 4), the trajectory cluster in fall came from the oceans more than
the one in winter. In winter, considerable air masses arriving at the site were also from
places near Nanjing (35%, cluster 2 in Fig. 13). Therefore, the aerosol kinds and
optical properties at the study site are characterized differently with different air
masses in the two seasons, which are further analyzed by their origins in SON and
DJF (Figs.12 and 14).

Figs. 12 and 14 show the main concentrations of particles and trace gases, the ratio of
$PM_{2.5}$ to $PM_{10}$, as well as the values of the aerosol optical properties of different
clusters during SON and DJF, respectively. Because $PM_{10}$ vary similarly to $PM_{2.5}$,
while $NO_x$ varies similarly to $NO_y$, we only show the variations of $PM_{2.5}$ and $NO_y$
with cluster here. Also, because AAC, SC and Bsp have good correlations with
particle concentrations (Zhuang et al., 2014) and Asp is greatly affected by relative
humidity (RH), we discuss the variation of SAE and SSA with cluster here.

In SON, the dominant air masses are from the East China Sea (passing through urban
agglomeration regions (cluster 3), and less-developed regions (cluster 2) of the YRD,
and northern continent away from Nanjing (cluster 4) (passing through oceans and



urban agglomeration regions). It is found that although air masses in cluster 3, cluster
4 and cluster 2 all pass through the oceans and have the same level of relative
humidity (RH), differences still exist among the clusters. The air masses have to cross
the urban agglomeration (from Shanghai to Nanjing) of YRD when they arrive in
Nanjing in cluster 3 but past less-developed regions (north Jiangsu Province) in
cluster 4 and cluster 2. In YRD, emissions of the aerosols and trace gases are much
stronger in urban agglomeration regions than those in other areas (Zhang et al., 2009;
Zhuang et al., 2013). It is also noticeable that concentrations of aerosols in cluster 4
are mostly lower, which may result from its avoidance from BTH regions, also a
megacities and urban agglomeration. In addition, air masses from the west of cluster 1
contain the highest concentrations of particulate matter, CO and $NO_y$, which may
result from crossing central China with high emission of CO according to MERRA
data (https://gmao.gsfc.nasa.gov/reanalysis/MERRA). Particulate matter and $NO_y$
mainly have the same sources as CO, and high concentrations of these aerosols are
also reflected in a high AOD according to the MISR data
(https://giovanni.gsfc.nasa.gov/giovanni). Zhuang et al. (2015) also suggested that
high emission occurred in central China. As for the ratio of $PM_{2.5}$ to $PM_{10}$, the ratio
represents the amount of particles deriving from secondary pollution progress
compared to those from primary pollution progress. Clusters 1-3 had relatively similar
ratios in SON, all over 60% except cluster 4, with the maximum of cluster 3, which
means particles deriving from secondary pollution progress in the three clusters have
a similar rate. $O_3$ concentrations among the four cluster were different. Despite



negative correlations of O₃ with its precursors and particles, the concentrations of $O_3$
in cluster 3 was higher than in cluster 4, as UV in cluster 3 was higher that in cluster 4.
The size of the aerosols in cluster 1 were finest (SAE is the largest in Fig. 12g),
because the other three clusters all pasted through oceans before arriving Nanjing,
with higher relative humidity (RH), making it easier for particles' hygroscopic growth.
SSA is also the largest in cluster 1, which means aerosols in cluster 1 are more
scattering.

In DJF, the air masses were from the places near Nanjing (cluster 2), northern
continent away from Nanjing (cluster 1), and northern continent away from Nanjing
passing through oceans and urban agglomeration regions (cluster 4). This is different
from that in SON. Therefore, besides what has been discussed of cluster 3 and cluster
4 in SON, it is found that air masses from cluster 1 and cluster 2 both account for over
30% of the total characteristics of the aerosol optical properties and are main sources
of pollutants in DJF (particles, CO, and $NO_x$ are higher in Fig.14). Air masses in
cluster 1 came from Shandong Province while those in cluster 2 came from areas
nearby. Particles and trace gases concentrations of cluster 2 are higher than those of
cluster 1 to some extent, which may result from the severer pollution in southern
YRD than in Shandong Province. The concentrations of $O_3$, similar to that in SON,
was affected by the UV (O₃ concentrations in cluster 2 is a little higher than that in
cluster 1 in Fig.14). The ratio of $PM_{2.5}$ to $PM_{10}$ of cluster 1 and cluster 2 are
approximately equal in DJF, over 70%. The size of aerosols in cluster 1 and 2 are



finer without passing through oceans, so SAE are larger (Fig.12g). Aerosols in cluster
1 are scatter to some extent compared to those in cluster 2.

**3.5 Case Study**
For further understanding of the causes for high pollutants episodes, especially high
particulate and $O_3$ episodes, we choose a typical episode from 2016 December 3-6 for
a detailed analysis.

Fig.15 (a) and (b) show that high $O_3$ concentrations (over 80 ppb) occurred on
December 4 with broad $O_3$ peaks (over 60 ppb) in the following days, while the
average $O_3$ during the cold seasons was 37.7 ppb. Though there is a lack of particulate
matter concentrations because of the instrument breakdown, we could see the high
concentrations of particulate matter from the relatively high EC value (over 500 $Mm^{-1}$)
and BC concentrations (over 6 $\mu g/m^3$) on December 4th, and both reach a maximum
on December 5th ($PM_{2.5}$ over 200 $\mu g/m^3$ and $PM_{10}$ over 300 $\mu g/m^3$), over 3 times of
the average concentrations. Besides, $NO_x$, $NO_y$, have reached high levels since
December 4th ($NO_x$ over 70 ppb and $NO_y$ over 100 ppb). It is also noticeable that
SSA has a relatively sharp decrease from December 4, especially on December 5
when particle concentrations were extremely high, representing that the ratio of $PM_{10}$
became higher. Meanwhile, a relatively sharp increase occurred in SAE, without any
obvious variation in AAE, though, which shows that scattering aerosols are the main



components. It is also found that this case occurred under calm conditions before the
passage of a cold front, which was at the front of a continental high pressure system
originating from Mongolia and sweeping over Nanjing (Fig.15 (c)), and the decrease
in temperature with high pressure system dominating eastern China were also
detected on December 6. Backward trajectories analysis for the past 96 hours (Fig.15
(d)) were conducted for December 5 at 8 pm for the maximum concentrations of $O_3$
on December 4 and particulate matter on December 5, which suggest that
predominant wind was just in time from the NW directions. Therefore, air masses
with high particles and $O_3$ concentrations would be transported to Nanjing, which
were also clearly detected in Nanjing during these days, such as the relatively high $O_3$
during nighttime on December 5 and 6. The highest $O_3$ on December 4 together with
high particles and primary pollutants $NO_x$ and $NO_y$ suggests a strong in situ
photochemical production in mixed regional plumes under the influence of high
pressure system. Guo et al. (2009) reported that the anticyclonic conditions, e.g.,
sunny weather and low wind velocities, are favorable for pollution accumulation and
$O_3$ production. Results in this case clearly demonstrate sub-regional transport of
primary and secondary air pollutants within the YRD region under such weather
system.
**4. Conclusion**
In this paper, an overview of particles and $O_3$ concentrations, together with trace gases,
during 2016 the cold seasons in urban Nanjing, China, has been presented based on



624 continuous measurements of aerosols concentrations and optical properties at the

625 Gulou site. The particles, $O_3$ and trace gases concentrations are comprehensively

626 characterized from perspectives of temporal variations, inter-species correlations,

627 trajectories analysis, and case studies based on weather data and Lagrangian

628 dispersion modeling.

630 Measurements show that hourly mean particle concentrations, including BC, $PM_{2.5}$,

631 and $PM_{10}$ at Gulou site, Nanjing, China, are $2.602 \pm 1.720$ μg/m$^3$, $58.2 \pm 36.8$ μg/m$^3$,

632 and $86.3 \pm 50.8$ μg/m$^3$, respectively, with ranges of 0.064-15.608 μg/m$^3$, 0.8-256.2

633 μg/m$^3$, and 1.1-343.4 μg/m$^3$, respectively. During the six months, 48 and 14 days

634 when $PM_{2.5}$ and $PM_{10}$, respectively, exceeded Class II NAAQS. Measurements also

635 showed that hourly mean $O_3$ concentrations in urban Nanjing ranged from 0.2 to

636 235.7 ppb, with average concentrations of $37.7 \pm 33.5$ ppb. There were 40 days excess

637 of $O_3$ during the period, suggesting a severe air pollution problem in the region.

639 The correlation analysis shows a negative $PM_{2.5}$–Vis correlation as well as RH, both

640 of which would promote the extinction coefficient. Negative $O_3$–$NO_y$ correlation

641 occurs when temperature is relatively low but the correlation becomes weaker when

642 temperature becomes higher. $PM_{2.5}$–$O_3$–T correlations reveal the formation of

643 secondary aerosols, especially fine particulate matter under high $O_3$ concentration and

644 temperature conditions, while BC–$O_3$–T correlations not. CO–$NO_y$–$O_3$ and $PM_{2.5}$–

645 $NO_y$–$O_3$ correlations suggest that a VOC-sensitive regime for photochemical



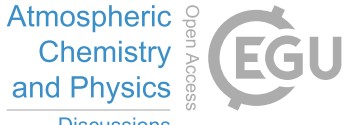

production of $O_3$ in urban Nanjing.

The backward trajectory analysis suggests that the prevailing winds in Nanjing were
from the north and east during the cold seasons in 2016. Air masses that are either
from the east without passing through the urban agglomeration and from northern
without crossing BTH regions were clean with low pollution concentrations. In
contrast, air masses from local regions were polluted in winter, suggesting a severe air
quality problem in YRD region. SAE and SSA were further studied, indicating that
particles from oceans were coarser and less scattering because the airmasses were
under high RH condition and less secondary pollutants were produced.

A case study for a typical high $O_3$ and $PM_{2.5}$ episode in December 2016 illustrates the
important influences of sub-regional transport of pollutants from strong source
regions and local synoptic weather on the episode. Stable conditions such as an
anticyclonic system make it easy for pollutants to accumulate in this region. Results
from this case reveal the mechanisms of sub-regional transport of primary and
secondary air pollutants within the YRD region.

Data availability. The automobile numbers and GDP are from http://www.njtj.gov.cn/.
Satellite CO data are available at: https://gmao.gsfc.nasa.gov/reanalysis/MERRA. The
aerosols AOD data are available at: https://giovanni.gsfc.nasa.gov/giovanni. The
Lagrangian dispersion model Hybrid Single-Particle Lagrangian Integrated Trajectory





(HYSPLIT) was supplied by NOAA: http://ready.arl.noaa.gov/HYSPLIT_traj.php.
The    meteorological    data    for    HYSPLIT    are    accessible    from
ftp://arlftp.arlhq.noaa.gov/pub/archives/gdas1.

**Acknowledgements:** This work was supported by the National Key R&D Program of China
(2017YFC0209803, 2014CB441203, 2016YFC0203303), the National Natural Science
Foundation of China (41675143, 91544230, 41621005). The authors would like to thank all
members in the AERC of Nanjing University for maintaining instruments.

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

Fig.4 Time series of particles from September 2016 to February 2017 at Gulou site.
Fig 5. Seasonal variations of (a) $O_3$, (b) $NO_x$, (c) CO, and (d) $NO_y$. The 10, 25, 50, 75, and 90%
percentile values of each are shown in black, and red markers represent the monthly averages.
Fig 6. 6-month mean diurnal variations of (a) trace gases and (b) UV (ultra-violate radiation) at





Gulou site from September 2016 to February 2017.
Fig 7. Scatter plots of (a) $O_3$-$NO_x$ color-coded with air temperature (T) and (b) $PM_{2.5}$-Vis
color-coded with relative humidity (RH).
Fig 8. Scatter plots of (a) $PM_{2.5}$-$O_3$ and (b) BC-$O_3$ color-coded with air temperature (T).
Fig 9. Scatter plots of (a) CO-$NO_x$, (b) $PM_{2.5}$-$NO_x$, and (c) BC-$NO_x$ color-coded with $O_3$.
Fig 10. Scatter plots of $PM_{2.5}$-$NO_x$ color-coded with $O_3$ during (a) daytime (9:00~ 17:00) and (b)
nighttime (0:00~ 6:00).
Fig 11. Clusters of 96 h back trajectories arriving at the study site at 100 m in 2016 fall.
Fig 12. The 10, 25, 50, 75, and 90% percentile values in each cluster of back trajectories in 2016
fall of (a) BC, (b) $PM_{2.5}$, (c) $PM_{2.5}$/$PM_{10}$, (d) CO, (e) $O_3$, (f) $NO_y$, (g) SAE, and (h) SSA. Black
markers represent the averages.
Fig 13. Clusters of 96 h back trajectories arriving at the study site at 100m in 2016 winter.
Fig 14. The 10, 25, 50, 75, and 90% percentile values in each cluster of back trajectories in 2016
winter of (a) BC, (b) $PM_{2.5}$, (c) $PM_{2.5}$/$PM_{10}$, (d) CO, (e) $O_3$, (f) $NO_y$, (g) SAE, and (h) SSA. Black
markers represent the averages.
Fig 15. Time series during December 3-6, 2016, for (a) $PM_{2.5}$, BC and $O_3$ with associated
meteological parameters, trace gases and (b) optical parameters. Red markers represent $O_3$ over
daily maximum average during winter. Weather charts on (c) 4th and (d) 5th December. (f) 96h
backward trajectories analysis ending at 1200 UTC on 5th December







**Table**

Table 1 Measurements at Gulou site.

| Measurement | | Instrument | Resolution |
|---|---|---|---|
| Meteorological parameters | T ($^\circ$C) | Thermo Instruments, THOM 1405-DF | |
| | P (atm) | Thermo Instruments, THOM 1405-DF | |
| | RH (%) | Thermo Instruments, THOM 1405-DF | |
| | Rainfall (mm) | | |
| | Vis (m) | Visibility Meter, GSN-1 | |
| | UV (W/m$^2$) | | |
| Particles | BC (ng/m$^3$) | Aethalometer, Model AE-31 | 1 ng/m$^3$ |
| | PM$_{2.5}$ ($\mu$g/m$^3$) | Thermo Instruments, THOM 1405-DF | 0.1$\mu$g/m$^3$ |
| | PM$_{10}$ ($\mu$g/m$^3$) | Thermo Instruments, THOM 1405-DF | 0.1$\mu$g/m$^3$ |
| Gaseous pollutant | CO (ppb) | Thermo Instruments, TEI 48i | 1 ppb |
| | NO$_x$ (ppb) | Thermo Instruments, TEI 42i | 0.4 ppb |
| | NO$_y$ (ppb) | Thermo Instruments, TEI 42iY | 0.4 ppb |
| | O$_3$ (ppb) | Thermo Instruments, TEI 49i | 0.01 ppb |
| Optical parameters | SC (Mm$^{-1}$) | Nephelometer, Aurora 3000 | 10$^{-3}$ Mm$^{-1}$ |
| | BSP (Mm$^{-1}$) | Nephelometer, Aurora 3000 | 10$^{-3}$ Mm$^{-1}$ |
| | AAC (Mm$^{-1}$) | Aethalometer, Model AE-31 | 10$^{-3}$ Mm$^{-1}$ |



Table 2 Statistics of general meteorological parameters at Gulou site for the 6-month period

September 2016~ February 2017.

| Month | Temp ($^\circ$C) | Pres (hPa) | RH (%) | Rainfall (mm) | Vis (km) | UV (W/m$^2$) |
|---|---|---|---|---|---|---|
| Sep | 24.88 | 996.97 | 69.41 | 2.34 | 11.84 | 10.36 |
| Oct | 18.37 | 1003.01 | 85.01 | 3.12 | 9.07 | 5.28 |
| Nov | 12.36 | 1007.87 | 77.15 | 1.19 | 8.99 | 5.67 |
| Dec | 8.74 | 1010.53 | 70.33 | 0.81 | 7.61 | 5.03 |
| Jan | 6.49 | 1010.89 | 70.65 | 0.59 | 9.23 | 4.94 |
| Feb | 7.72 | 1009.65 | 59.99 | 0.45 | 10.24 | 7.04 |







Table3.a Nephelometer correction factors for angular nonidealities. Wavelengths for Aurora 3000
are 450 nm (B), 525 nm (G), and 635 nm (R), respectively.

| | Midpoint±half range of calculated correction factors | | | | | |
|---|---|---|---|---|---|---|
| | total scatter | | | back scatter | | |
| wavelength | B | G | R | B | G | R |
| | 1.37±0.29 | 1.38±0.31 | 1.36±0.29 | 0.963±0.040 | 0.971±0.047 | 0.968±0.043 |



Table3.b Correction factors for total scatter as function of Ångström exponent: $C_{ts} = a + b \cdot \alpha_{ts}^{*}$ .
For correction of scattering coefficients for the blue (B) wavelength the Ångström exponent
calculated from uncorrected scattering coefficients of blue and green (B/G) is used. At the
wavelength G and R Ångström exponents at the wavelength pairs B/R and G/R are used,
respectively.

| wavelength | B | | G | | R | |
|---|---|---|---|---|---|---|
| Ångström exponents | $\alpha_{ts}^{*}(B/G)$ | | $\alpha_{ts}^{*}(B/R)$ | | $\alpha_{ts}^{*}(G/R)$ | |
| parameters | a | b | a | b | a | b |
| | 1.455 | −0.189 | 1.434 | −0.176 | 1.403 | −0.156 |



Table 4 Statistics of general aerosol optical parameters at Gulou for the 6-month period September
2016–February 2017.

| Factors | Mean ± STD | Median | Maximum | Minimum |
|---|---|---|---|---|
| 550 nm AAC (Mm$^{-1}$) | 23.741 ± 15.557 | 20.568 | 127.167 | 0.994 |
| 550 nm SC (Mm$^{-1}$) | 349.502 ± 235.291 | 300.901 | 1873.620 | 17.436 |
| 550 nm BSp (Mm$^{-1}$) | 35.469 ± 21.488 | 31.637 | 161.958 | 1.383 |
| 550 nm EC (Mm$^{-1}$) | 373.536 ± 247.877 | 323.070 | 1947.900 | 22.198 |
| 550 nm SSA | 0.929 ± 0.028 | 0.933 | 0.985 | 0.743 |
| 550 nm Asp | 0.645 ± 0.052 | 0.648 | 0.902 | 0.386 |





| | | | |
|---|---|---|---|
| 470/660 nm AAE | 1.600 ± 0.175 | 1.611 | 2.822 | 0.059 |
| 450/635 nm SAE | 1.192 ± 0.288 | 1.192 | 2.159 | 0.256 |

Table5. Statistics of the three particles during the study period at Gulou site, Nanjing, China

| | Mean ± STD | Median | Maximum | Minimum |
|---|---|---|---|---|
| BC (μg/m³) | 2.602 ± 1.720 | 2.241 | 15.609 | 0.064 |
| PM$_{2.5}$ (μg/m³) | 58.2 ± 36.8 | 49.3 | 256.2 | 0.8 |
| PM$_{10}$ (μg/m³) | 86.3 ± 50.8 | 76.7 | 343.4 | 1.1 |



Table6. Statistics of trace gases during the study period

| | Mean ± STD | Median | Maximum | Minimum |
|---|---|---|---|---|
| CO (ppb) | 851 ± 384 | 765 | 2852 | 176 |
| O$_3$ (ppb) | 37.7 ± 33.5 | 31.0 | 235.7 | 0.2 |
| NO$_x$ (ppb) | 23.5 ± 14.7 | 19.5 | 80.0 | 2.7 |
| NO$_y$ (ppb) | 32.8 ± 22.3 | 26.7 | 158.4 | 3.6 |



Table7. Statistics of maximum and number of exceedances of O$_3$ and PM$_{2.5}$ compared with the

National Ambient Air Quality Standards in China.

| Aerosol | Mean ± STD (μg/m³) | Max (μg/m³) | N.o.E. |
|---|---|---|---|
| PM$_{2.5}$ | 58.2 ± 36.8 | 256.2 | 48 |
| PM$_{10}$ | 86.3 ± 50.8 | 343.4 | 14 |
| O$_3$ | 80.8 ± 71.8 | 235.7 | 37 |

N.o.E. of PM$_{2.5}$ accounts for days with 24 h average over 75 μg/m³. N.o.E. of PM$_{10}$ accounts for days
with 24 h average over 150 μg/m³. N.o.E of O$_3$ accounts for days with maximum 8 h average exceed
160 μg/m³.



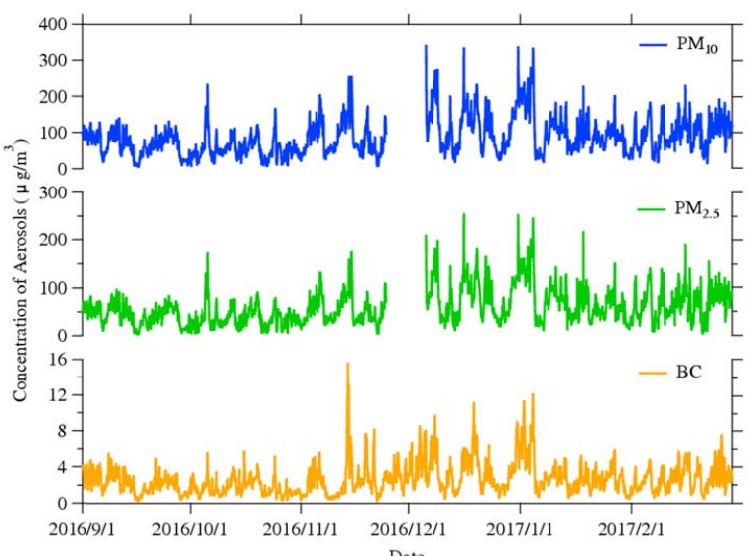


Fig 1. Time series of the concentrations of PM$_{10}$, PM$_{2.5}$, and BC from September 2016 to February

2017 at Gulou site, Nanjing, China.






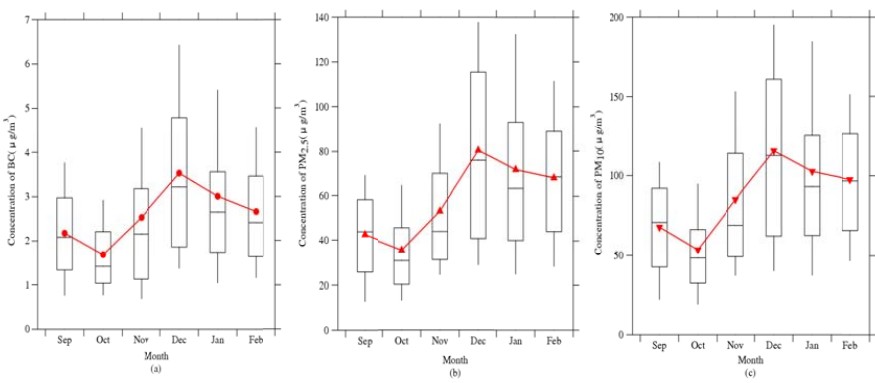


Fig 2. Seasonal variations of (a) BC, (b) PM$_{2.5}$, and (c) PM$_{10}$. Red markers represent the monthly

averages at Gulou site, Nanjing, China.




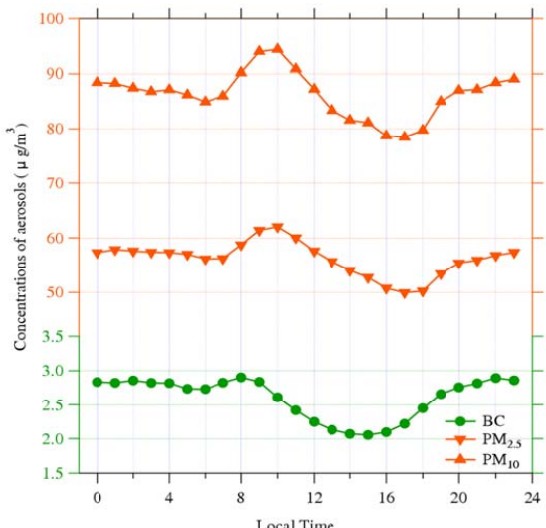


976 Fig 3. 6-month mean diurnal variations of BC, PM$_{2.5}$, and PM$_{10}$ at Gulou site, Nanjing, China

977        from September 2016 to February 2017.




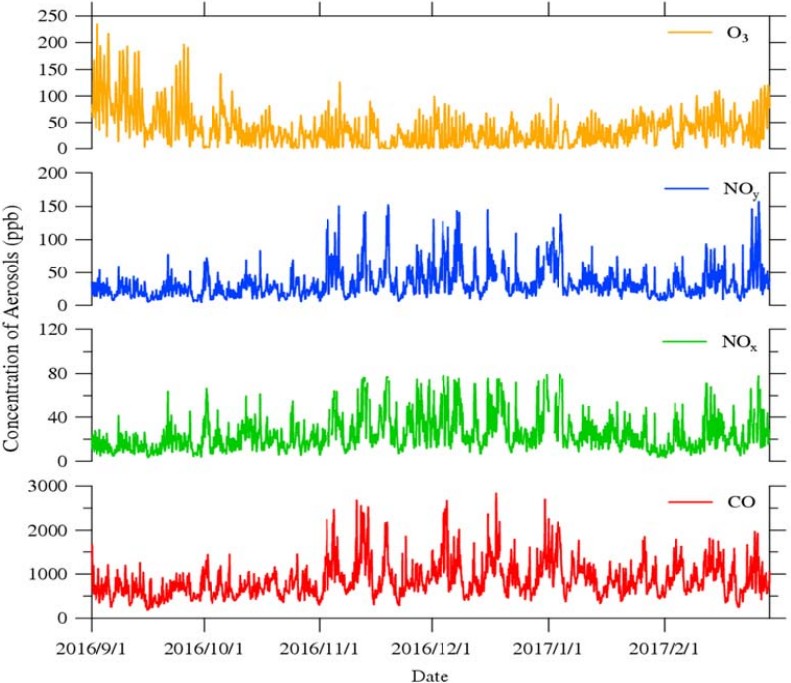


Fig.4 Time series of particles from September 2016 to February 2017 at Gulou site.




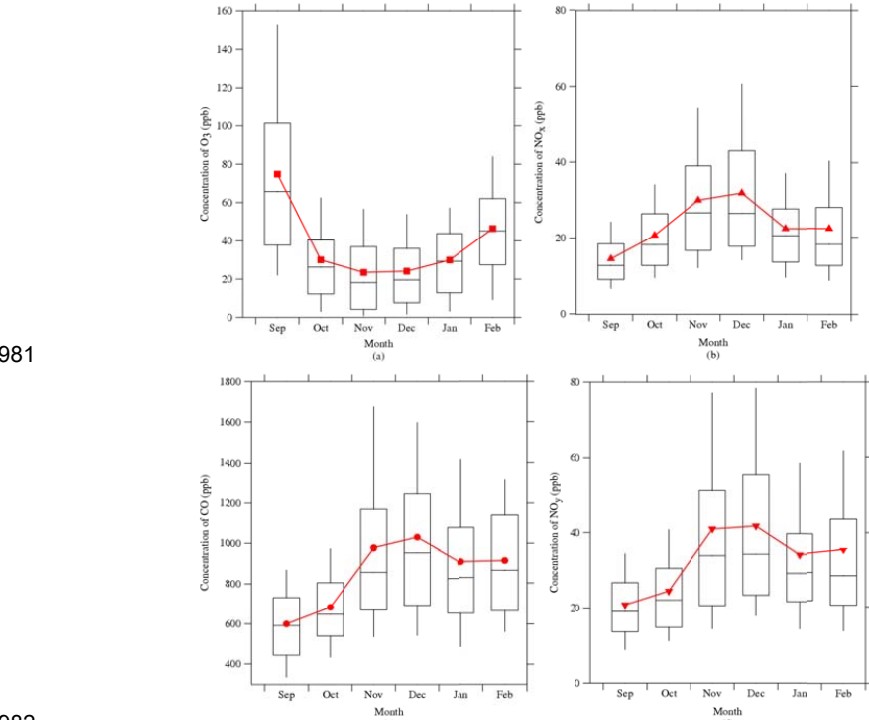



Fig 5. Seasonal variations of (a) O$_3$, (b) NO$_x$, (c) CO, and (d) NO$_y$. The 10, 25, 50, 75, and 90%

percentile values of each are shown in black, and red markers represent the monthly averages.





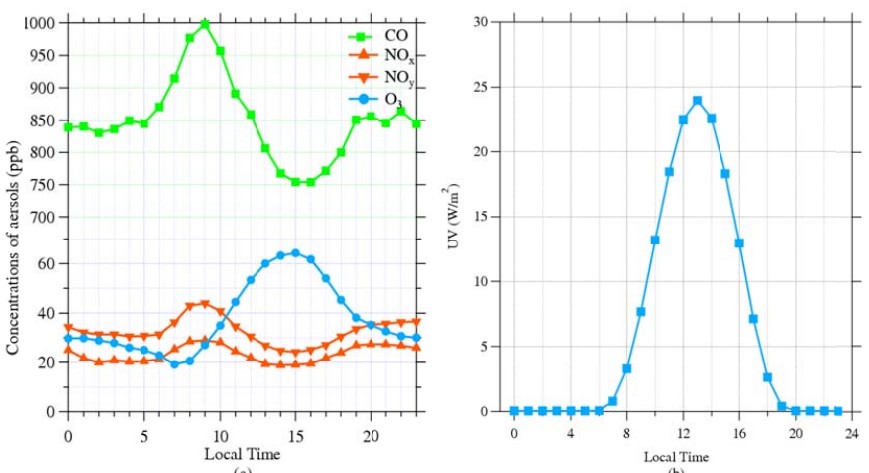


Fig 6. 6-month mean diurnal variations of (a) trace gases and (b) UV (ultra-violet radiation) at

Gulou site from September 2016 to February 2017




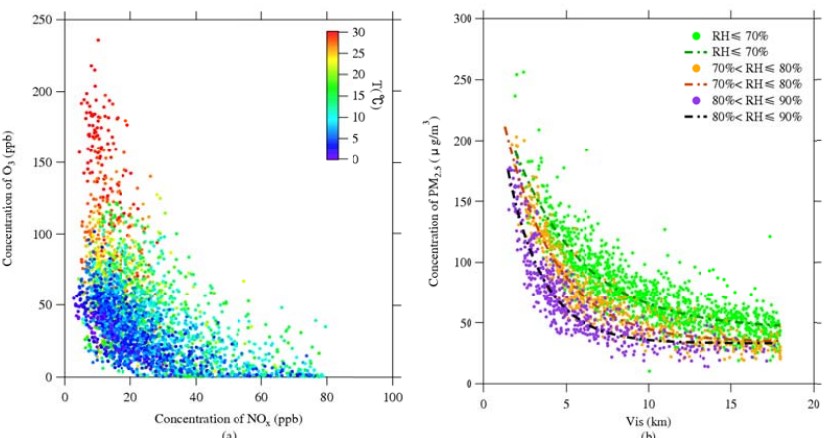


Fig 7. Scatter plots of (a) $O_3$-$NO_x$ color-coded with air temperature (T) and (b) $PM_{2.5}$-Vis

color-coded with relative humidity (RH).



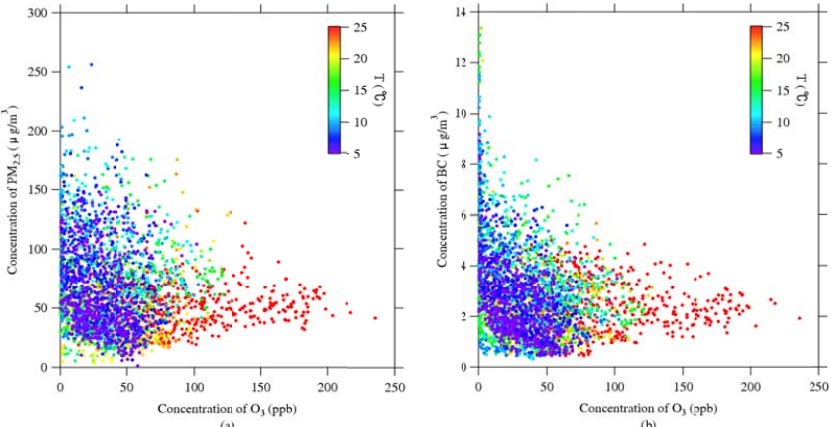


Fig 8. Scatter plots of (a) PM$_{2.5}$-O$_3$ and (b) BC-O$_3$ color-coded with air temperature (T).





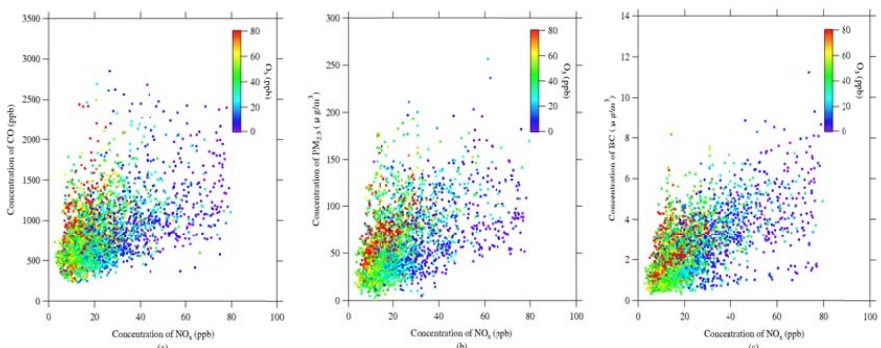


Fig 9. Scatter plots of (a) CO-NO$_x$, (b) PM$_{2.5}$-NO$_x$, and (c) BC-NO$_x$ color-coded with O$_3$.





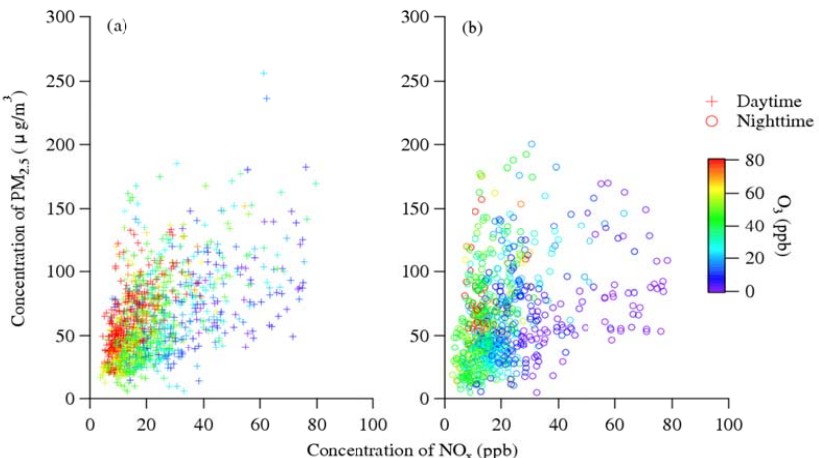


001     Fig 10. Scatter plots of PM$_{2.5}$-NO$_x$ color-coded with O$_3$ during (a) daytime (9:00~ 17:00) and (b)

nighttime (0:00~ 6:00).





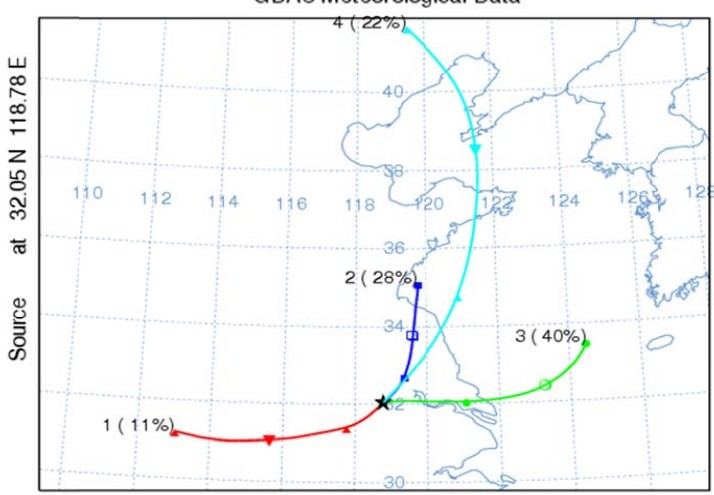


Fig 11. Clusters of 96 h back trajectories arriving at the study site at 100 m in 2016 fall.







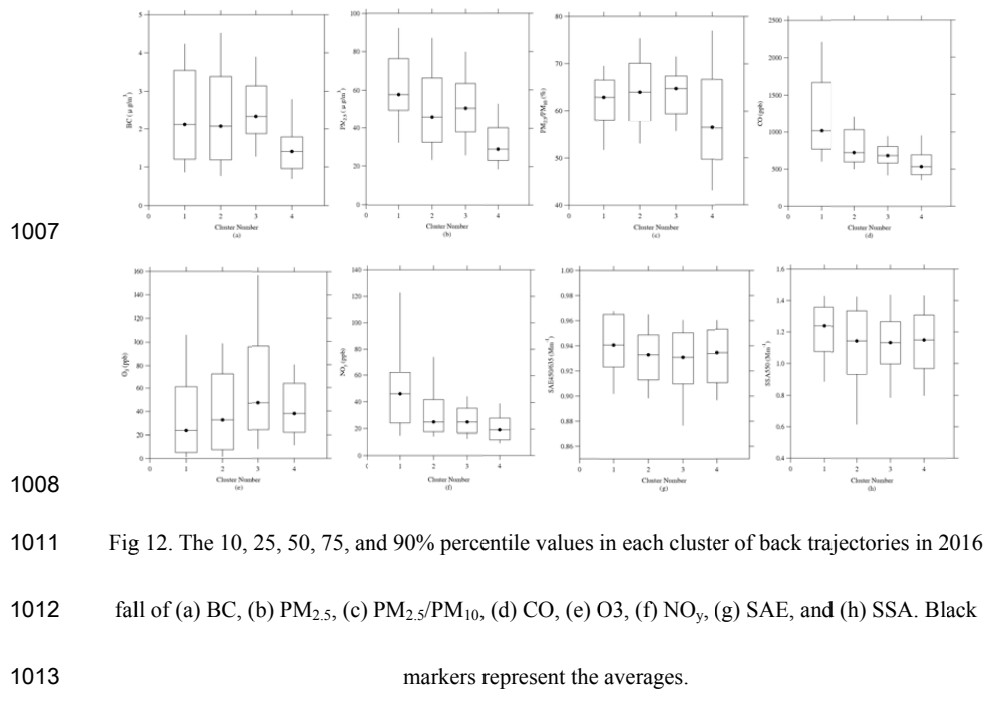



Fig 12. The 10, 25, 50, 75, and 90% percentile values in each cluster of back trajectories in 2016
fall of (a) BC, (b) PM$_{2.5}$, (c) PM$_{2.5}$/PM$_{10}$, (d) CO, (e) O3, (f) NO$_y$, (g) SAE, and (h) SSA. Black

markers represent the averages.

1012





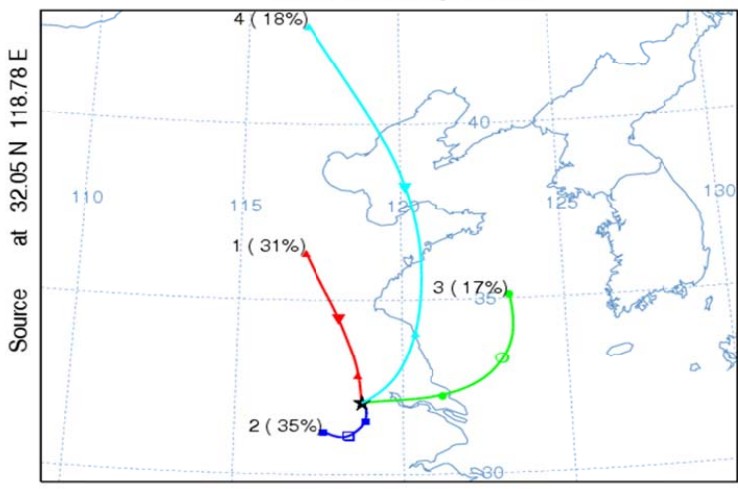

1013

1014 Fig 13. Clusters of 96 h back trajectories arriving at the study site at 100m in 2016 winter.



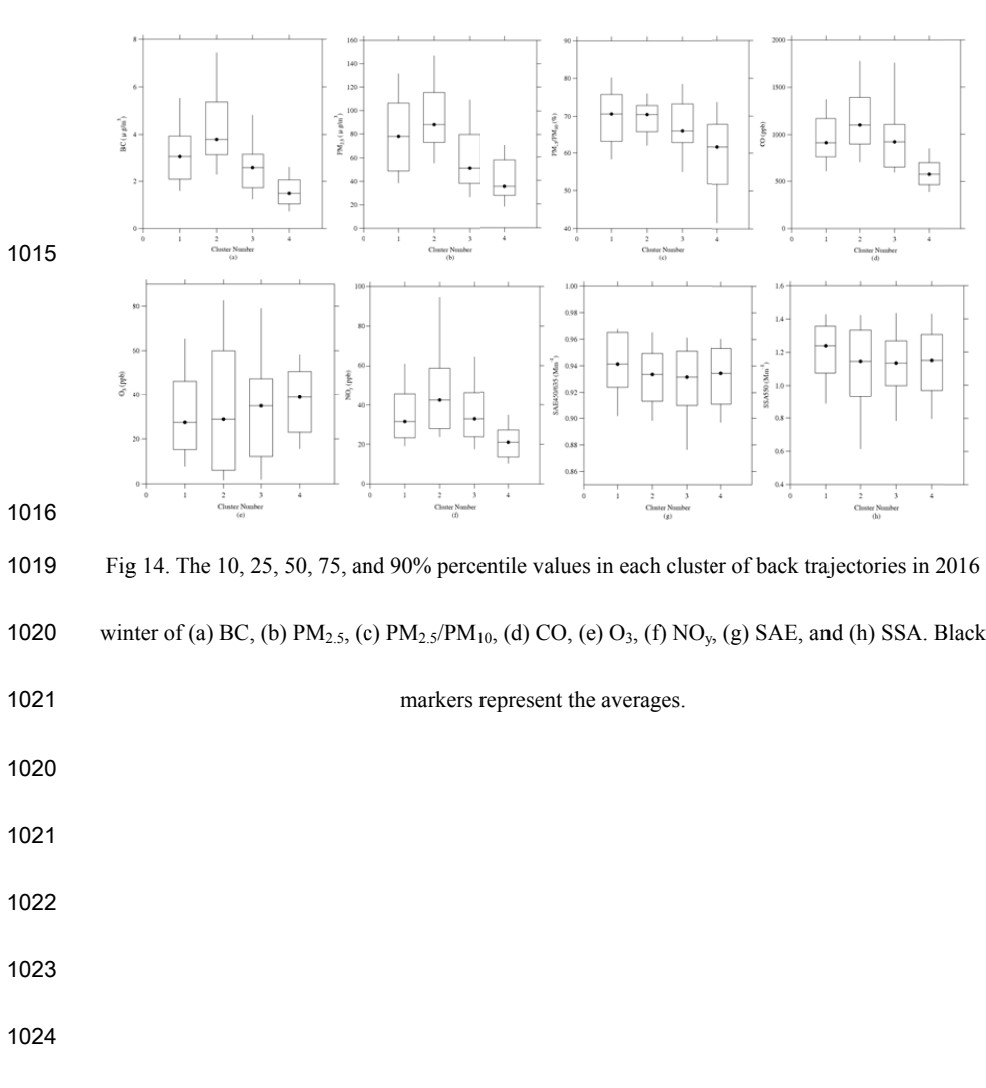

Fig 14. The 10, 25, 50, 75, and 90% percentile values in each cluster of back trajectories in 2016

winter of (a) BC, (b) PM$_{2.5}$, (c) PM$_{2.5}$/PM$_{10}$, (d) CO, (e) O$_3$, (f) NO$_y$, (g) SAE, and (h) SSA. Black

markers represent the averages.














(c)                              (d)

(f)





1034         Fig 15. Time series during December 3-6, 2016, for (a) PM2.5, BC and O3 with associated

meteological parameters, trace gases and (b) optical parameters. Red markers represent O3 over
daily maximum average during winter. Weather charts on (c) 4th and (d) 5th December. (f) 96h

backward trajectories analysis ending at 1200 UTC on 5th December.
