# Peer review of "Characteristics of ozone and particles in the near-surface"

_Atmospheric Chemistry and Physics, 2018_

## Referee Comment (RC1) · Anonymous Referee #2 · 5 Dec 2018

A major revision of the MS must be made. Reconsideration of the MS is only possible pending the responses from the authors to the points listed below. The MS reports the observational data but barely digs enough into it, let alone a sufficient and reasonable discussion without conceptual mistakes. Moreover, the MS is not comfortably readable and lacks brevity. There are quite a few grammatical errors to be corrected. It would also be better if the language could be polished in the revision.

Main points: 1. The structure of the introduction apparently lacks logic organization. Even more, major scientific issues the MS to be addressed are not clearly stated. 2. A detailed description of the environment where all instruments are installed should be

given in section 2.1. How about the drying system upstream AE-31 and Aurora-3000? The instruments used to measure trace gases should be at least briefly described, instead of having not even a single word on that. 3. The SC2006 is adopted in this study to correct the systematic biases inherent in the principle of AE-31. What are the parameters used in your procedure? How about the values of your correction factors? The description needs to be more specific. 4. The truncation correction of Aurora-3000 is based on Mie calculations. If I understand it correctly, Mie calculation is not performed in this study, since obviously there is no measurement of particle number size distribution. Instead, correction parameters are directly taken from the literature in this study. How much uncertainty might be introduced to scattering coefficients due to the choice of the correction factors? 5. HYSPLIT model is driven by NCEP data with a temporal resolution of 6 hours and a spatial resolution of 2.5 degrees in this study. I doubt that the resolution is adequate for carrying a simulation of near surface transport process. 6. Page 14, Line 283-284, particles especially sub-micron particles could hardly be removed from the atmosphere by rain droplets. It is strong wind before the rain that sweeps them out. 7. Page 18, Line 385-386, the author states that the diurnal pattern of NOx is mainly governed by photochemical processes and meteorology. However, emission is a key factor that should not be ignored. 8. Page 22, Line 454-460, the concept about fog is completely wrong. Fog only occurs above 100% RH, though droplets can exist below 100% due to the hygroscopic growth of particles. 9. Page 22, Line 467-471, the existence of aerosols might affect solar radiation to some extent and thus ozone photochemical production (not always to a measurable amount). However, the main reason for the observed variation of ozone should not be attributed to aerosols. The author tempts to build a relationship between aerosols and ozone, but I find the analysis of data and deduction not robust and even incorrect, just like here and discussions elsewhere in the MS, e.g, Page 24, Line 503-504. 10. Page 23, Line 491-499, the author draws a conclusion that ozone photochemical production is VOC-limited by using CO/particle-O3-NOx relationship. I find it very unconvincing.

Minor points: 1. The full form should be given for the abbreviation of BTH in the ab-

stract. 2. The abbreviation of several aerosol optical parameters such as AAC and SC is not common. It would be better to follow the convention in the community. 3. Page 13, Line 270-274, the sentence beginning with 'For example' is supposed to illustrate the point brought forward by the sentence before it. However, I find their connection rather confusing.

---

## Referee Comment (RC2) · Anonymous Referee #3 · 21 Dec 2018

The paper written by Chen et al., performed the continuous measurements of particles and trace gases in Nanjing during cold seasons. Although the interaction of atmospheric components (e.g., trace gases, aerosols) and meteorological conditions has been analyzed, the originality should be addressed especially in abstract before publication. Besides, the paper still suffered from many minor flaws throughout the manuscript. Thus, I suggest this paper could be published after revising the minor errors. The detailed suggestions are as follows: 1. It was well documented that the air pollutants were closely linked to the weather system and meteorological conditions. (Line 32) The author only revealed the important effects of weather system and human activities on the environment in the YRD region, which has been investigated by

many previous studies. The originality was not addressed in the manuscript. In my opinion, the abstract should be rewritten to stress the new contribution of this paper to atmospheric chemistry rather than reporting the pollution level simply. 2. Line 71, the author said observation-based studies of particles were relatively limited. I think it was very subjective because there were hundreds of observation-based studies about the aerosol particles in the past decades. Meanwhile, in line 75, the author said there were only very limited studies of O3 in the urban of YRD. Actually, the O3 concentration has been widely monitored in YRD because it was one of the most important gaseous pollutants in YRD. I think the author should review a large amount of papers before writing this paper. 3. Line 108-112, the author should highlight the objective of the present study. In addition, the sentence between line 110 and line 112 should be replaced by the environmental implication of the research. 4. Line 123, the instruments used to monitor the gaseous pollutants such as O3 should be added in the methods. Additionally, NOy generally consisted of a large of N-bearing gaseous pollutants. The detailed NOy species should be introduced in this part. 5. Line 263-264, the author did not show the variation trend of BC, PM10, and PM2.5. Furthermore, how do you know the sources of these pollutants shared the similar sources? The relevant references were also missing. Line 265, what does transport emission mean? 6. Line 272-274, the author said the high loadings of particulate matter in early October was mainly due to the increase in aerosol concentrations with high scatter coefficient (SC). I do not understand the association between PM concentration and the aerosol concentrations with high SC. Please explain the reasons in details. 7. Line 284-286, Nanjing is located in Southeast China. The combustion of fossil fuels for domestic heating is not common in the winter of Nanjing. I do not understand why the increased anthropogenic emission of fossil fuels in the winter of Nanjing contributed to the high aerosol loadings. 8. Line 294, the diurnal variation of BC concentration was generally associated with the vehicle volume. I am very curious about the higher BC levels during 8-11 pm. I think Nanjing showed the higher vehicle volume during 5-8 pm. The author should explain the unusual characteristics. 9. Line 336, the author inferred that the BC and CO in the

atmosphere were mainly originated from biomass burning. The fire point data should added to demonstrate the potential source of BC and CO. 10. Line 495-496, what does the sentence mean? The author should point out the relationship between CO and ozone production. 11. The conclusion should be condensed and stress the new contribution to the atmospheric chemistry. 12. There are many grammar and format errors throughout the paper. I suggest the author should revise all of these minor flaws from words to words carefully.

---

## Author Comment (AC1) · 15 Feb 2019

To Editors and Anonymous Referee #3:

Dear editors and reviewers:

Thank you very much for dedicating time to reviewing the manuscript and providing us the important comments and suggestions on our study. We have learned a lot from your advice and made great efforts to improve the manuscript accordingly. A carefully point by point response to your comments has been listed below which we hope meet with approval. The revised details can be referred to the new version of the manuscript

in the supplement.

Anonymous Referee #3

The paper written by Chen et al., performed the continuous measurements of particles and trace gases in Nanjing during cold seasons. Although the interaction of atmospheric components (e.g., trace gases, aerosols) and meteorological conditions has been analyzed, the originality should be addressed especially in abstract before publication. Besides, the paper still suffered from many minor flaws throughout the manuscript. Thus, I suggest this paper could be published after revising the minor errors.

The detailed suggestions are as follows:

1. It was well documented that the air pollutants were closely linked to the weather system and meteorological conditions. (Line 32) The author only revealed the important effects of weather system and human activities on the environment in the YRD region, which has been investigated by many previous studies. The originality was not addressed in the manuscript. In my opinion, the abstract should be rewritten to stress the new contribution of this paper to atmospheric chemistry rather than reporting the pollution level simply.

R:

We sincerely thanks for your comments. In the revised manuscript, the authors stress the originality of the study.

1. Indeed, some researches on the air pollutants related to weather system and human activities have been carried out in most sites of YRD recently. However, previous studies using observation data in Nanjing often concentrated on characteristics of one of the particles, such as BC and carbonaceous aerosols (e.g., Zhuang et al., 2014), or PMs (e.g., Deng et al., 2011; Shen et al., 2014), or ozone and its precursors (e.g., Tu et al., 2007; Wang et al., 2008; An et al., 2015). Thus, it is necessary to achieve a relatively comprehensive understanding of the air pollution problem directly through analysis of various species. In addition, most of them described the temporal and spatial distributions of concentrations, and the influence of meteorological effects. In this study, discussion of aerosols characteristics, especially particles, is not limited to the concentrations but taking optical properties into consideration as well. Moreover, most of them lay less emphasis on the inter-species correlations and the combined effects of more than one pollutant, especially the possible underlying chemical progress, during severe pollution episodes except Ding et al. (2013b), who described the characteristics of O3 and PM2.5 with near-surface observation data in rural area of Nanjing. As implied in Zhang et al. (2012), aerosols are complicated in compositions and spatial distributions especially in fast developing regions with intense human activities (such as Nanjing). Thus, differences of the aerosol characteristics, for instance, concentrations as well as optical properties, might exist to degrees among the sites located in different parts of Nanjing with different land use. Additionally, a better understanding of spatial and temporal variations of pollutants can contribute to the adoption of effective measures to reduce air pollution on the urban scale. Therefore, it's necessary to investigate the characteristics of air pollutants in urban area of west YRD.

2. To make a better insight of the correlation and interaction between particles and ozone (the two main pollutants) through observation data, this study further identifies the influence of associated affecting factors, including UV radiation, temperature, and precursor's concentrations on the interaction (Section 3.3). Most of previous studies present the findings from various numerical models (e.g., Li et al., 2005; Bian et al., 2007; Deng et al., 2010; Li et al., 2011; Li et al., 2018, etc.). However, only a few studies discussed the correlation based on observation. In Nanjing, Ding et al. (2013b) described a correlation between PM2.5 and O3. But only temperature is regarded as an affecting factor. Thus, it is believed that our study would contribute to a more comprehensive understanding of the underlying mechanisms from observation.

3. Back to the site, the site is located in the city center, one of the highly residential areas of Nanjing, with concentrated human activities with residential areas, schools, institutions and business districts, and the main road of urban transportation around. Therefore, the results could suggest the characteristics and interactions of pollutants in the urban region very well. Also, the results could further imply the effects of the urban underlying surface and human activities to degrees. Besides, as a typical urban area, the results in this study would probably bring new knowledge of aerosol characteristics, like the pollution level variation in different years and different regions through comparison with previous studies based on observation and numerical simulations.

Overall, this manuscript presents more comprehensive, systematic and deeper analysis on main pollutants like particles and ozone in urban area of west YRD. Results further indicate the characteristics of the particles and trace gases and reveal the possible chemistry process and interactions among different species and meteorological variables in west YRD. And they are also advantageous to improve the understanding of the detailed variations (seasonal, monthly, and diurnal) and its effects in east regions of China.

According to your comments, questions and suggestions, not only the abstract, but the introduction, discussion and conclusion have also been rephrased carefully. The originality (listed above) and finding(s) of this study have been refined in better ways of expression. Details can be found in the revised manuscript.

**Supplement:**

[revised manuscript text omitted]
 come from the local region were from the places near Nanjing (cluster 2), north-west areasnorthern continent away from Nanjing (cluster 1), and northern regionsnorthern continent away far from Nanjing passing through oceans and urban agglomeration regions (cluster 4). Air masses from cluster 1 and cluster 2 both account for over 30% of the total aerosol characteristics and are more polluted with relatively high levels of particles, CO, and NO$_x$. Air masses in cluster 1 come from Shandong Province while those in cluster 2 come from local areas. Particles and trace gases concentrations of cluster 2 are higher than those of cluster 1 to some extent, implying the severer air pollution problem in YRD region. This is different from that in SON. Therefore, besides what has been discussed of cluster 3 and cluster 4 in SON, it is found that air masses from cluster 1 and cluster 2 both account for over 30% of the total characteristics of the

The concentrations of $O_3$, similar to that in SON, is affected by radiation besides precursors levels. Thus, $O_3$ concentration in cluster 2 is a little higher than that in cluster 1.

The ratios of $PM_{2.5}$ to $PM_{10}$ of cluster 1 and cluster 2 are approximately equal in DJF, over

70%. The size of aerosols in cluster 1 and 2 are coarser, however, probably due to the higher RH

(over 65%). Aerosols in cluster

1 are scatter to some extent compared to those in cluster 2. The trajectories of cluster 3 and cluster

4 are analogous to those in SON, respectively, but more polluted, probably due to more emissions in DJF especially in north China and weaker flow from ocean in DJF.

**3.5 Case Study**

For further understanding of the causes for high pollutants episodes, especially high particulate and

$O_3$ episodes, detailed analysis of a typical episode from 2016 December 3-6 is presented in this section.

Fig.15 (a) and (b) show that high $O_3$ concentrations (over 80 ppb) occurred on December 4 with broad $O_3$ peaks (over 60 ppb) in the following days, while the average $O_3$ during the cold seasons was 37.7 ppb. Though there is a lack of particulate matter concentrations because of the instrument breakdown,  high concentrations of PMs might possibly occur referring to  the relatively high $\sigma_e$  value (over 500 $Mm^{-1}$) and BC concentrations (over 6

$\mu g/m^3$) on December 4th Both PMs reach a maximum on December 5th ($PM_{2.5}$ over 200

$\mu g/m^3$ and $PM_{10}$ over 300 $\mu g/m^3$), over 3 times of the average concentrations. Besides, $NO_x$, $NO_y$, have reached high levels since December 4th ($NO_x$ over 70 ppb and $NO_y$ over 100 ppb). It is also noticeable that $\omega_0$  has a relatively sharp decrease from December 4th, especially on December

5th when particle concentrations were extremely high,  probably suggesting that the ratio of $PM_{10}$ became higher. Meanwhile, a relatively sharp increase occurred in $\alpha_{ts}$ , without any obvious variation in $\alpha_a$ , though, implying that scattering aerosols could take the leading role during this episode. It is also found that this case occurred under calm conditions before the passage of a cold front, which was  in the front of a continental high pressure system originating from Mongolia and sweeping over Nanjing (Fig.15 (c)).

And the decrease in temperature with high pressure system dominating eastern China  wes also detected on December 6th. Backward  trajectory analysis for the past 96 hours (Fig.15

(d))  was conducted  from December 5th at 20:00 LT,  including the maximum concentrations of $O_3$ on December 4th and PMs on December 5th  
[revised manuscript text omitted]

---

## Author Response (AR1)

**To Editors and Anonymous Referee #2 and #3:**

Dear editors and reviewers:

Thank you very much for dedicating time to reviewing the manuscript and providing us the important comments and suggestions on our study. We have learned a lot from your advice and made great efforts to improve the manuscript accordingly. A carefully point by point response to your comments has been listed below which we hope meet with approval. The revised details can be referred to the new version of the manuscript.

**Relevant changes of the revised manuscript (marked with traces) as well as the change list are also enclosed in the last part of this document.**

**Anonymous Referee #2**

A major revision of the MS must be made. Reconsideration of the MS is only possible pending the responses from the authors to the points listed below. The MS reports the observational data but barely digs enough into it, let alone a sufficient and reasonable discussion without conceptual mistakes. Moreover, the MS is not comfortably readable and lacks brevity. There are quite a few grammatical errors to be corrected. It would also be better if the language could be polished in the revision.

**R:** We sincerely thanks for pointing out the problem of manuscript's analysis and writings. First, according to your suggestions, the authors conduct a more detailed analysis and discussion on the observational data. And we have also checked and corrected the all confusing statements in the manuscript. For example, we describe the similar role CO play in ozone production as volatile organic compounds (VOCs) and the criterion for VOC/$NO_x$ sensitive region in the revised manuscript after a comprehensive study of related work, making the deduction of the VOC-limited region through the CO-$NO_x$-$O_3$ correlation in this study more convincing. Also, for a sufficient use of observation data, like the aerosol optical properties, we have further analyzed the optical properties data to some extent for a better understanding of particle characteristics, such as its size and light extinction effects. It would contribute to the analysis of aerosols characteristics. Moreover, the manuscript has been rephrased significantly and shortened in necessarily throughout the whole text. Most parts of the manuscript have been shortened, especially for the Sections 2.2, 3.1, and 3.2. For example, the calculation of the aerosol optical properties and truncation correction of Aurora-3000 (Section 2.2), which have been stated clearly in previous studies (e.g., *Zhuang et al., 2015, 2017*; *Anderson and Ogren,1998*; *Müller et al., 2011*, *etc.*) have been rephrased to a briefer but more legible version. More details could be found in the revised manuscript, and it's believed that the revised version of the manuscript is much clearer and more readable.

With regard to your comments, questions and suggestions, the manuscript has been rephrased throughout the whole text. The finding(s) of this study have also been refined in better ways of expression, which could be found in most parts of the revised manuscript, including in the sections of Abstract, Introduction, Discussions, as well as Conclusion. Details can be found in the revised manuscript.

**References:**

Zhuang, B. L., Wang, T. J., Liu, J., Ma, Y., Yin, C. Q., Li, S., Xie, M., Han, Y., Zhu, J. L., Yang, X. Q., Fu, C. B.: Absorption coefficient of urban aerosol in Nanjing, west Yangtze River Delta, China, Atmos. Chem. Phys., 15, 13633–13646, doi:10.5194/acp-15-13633-2015, 2015.

Zhuang, B. L., Wang, T. J., Liu, J., Li, S., Xie, M., Han, Y., Chen, P. L., Hu, Q. D., Yang X.Q., Fu, C. B., Zhu, J. L.: The surface aerosol optical properties in the urban area of Nanjing, west GTH River Delta, China. Atmos. Chem. Phys., 17, 1143–1160, doi:10.5194/acp-17-1143-2017, 2017.

Anderson, T. L., Ogren, J. A.: Determining aerosol radiative properties using the TSI 3563 integrating nephelometer, Aerosol Sci. Tech., 29, 57–69, 1998.

Müller, T., Laborde, M., Kassell, G., Wiedensohler, A.: Design and performance of a three-wavelength LED-based total scatter and backscatter integrating nephelometer, Atmos. Meas. Tech., 4, 1291–1303, doi:10.5194/amt-4-1291-2011, 2011.

Main points:

1. The structure of the introduction apparently lacks logic organization. Even more, major scientific issues the MS to be addressed are not clearly stated.

**R:** According to your suggestions, the structure of the introduction has been reorganized. It is believed to be more readable and easier for readers to grasp the major scientific issues of the study. Please refer to the revised manuscript for more details.

2. A detailed description of the environment where all instruments are installed should be given in section 2.1. How about the drying system upstream AE-31 and Aurora-3000? The instruments used to measure trace gases should be at least briefly described, instead of having not even a single word on that.

**R:** Thank you for your suggestions and question. Section 2.1 has been extended to degrees. Description of the environment where all instruments are installed has been included in the revised manuscript. For the drying system, there is no heater for AE-31, which is similar to the settings in other sites (e.g., *Wu et al., 2012*; *Wu et al., 2013*; *Gong et al., 2015, etc.*). Both external and internal heaters are equipped for Aurora-3000. However, the internal heater has been turned off during the study period because RH in the tube is mostly lower than 50% in this period. Corresponding statements on settings of AE-31 and Aurora-3000 have also been included in the revised manuscript. For the instruments of the trace gases, more detailed description can be found in the revised manuscript.

**References:**

Wu, Y. F., Zhang, R. J., Pu, Y. F., Zhang, L. M., Ho, K. F., Fu, C.B.: Aerosol optical properties observed at a semi-arid rural site in northeastern China, Aerosol Air Qual. Res., 12, 503–514, 2012.

Wu, D., Wu, C., Liao, B., Chen, H., Wu, M., Li, F., Tan, H., Deng, T., Li, H., Jiang, D., Yu, J. Z.: Black carbon over the South China Sea and in various continental locations in South China, Atmos. Chem.
Phys., 13, 12257–12270, doi:10.5194/acp-13-12257-2013, 2013.
Gong, W., Zhang, M., Han, G., Ma, X., Zhu, Z.: An investigation of aerosol scattering and absorption
properties in Wuhan, Central China, Atmosphere, 6, 503–520, 2015

3.  The SC2006 is adopted in this study to correct the systematic biases inherent in the principle
of AE-31. What are the parameters used in your procedure? How about the values of your
correction factors? The description needs to be more specific.

**R:** Thanks for your question. Previous investigation indicated that both Weingartner corrected (WC2003
for short, hereinafter) and Schmid corrected (SC2006 for short, hereinafter) absorptions show good
agreements with the one from the Multi-Angle Absorption Photometer (*Collaud Coen et al., 2010*).
Therefore, we have applied several correction algorithms to calculate the aerosol absorption coefficient
according to SC2006, WC2003, and indirect calculation (IDC). And the aerosol optical properties and
certain parameters used in the correction procedures are based on our observation data and previous work
(*Wu et al., 2009*; *Wu et al., 2013*). Results showed that corrected $\sigma a$ $\sigma_a$ at 532 nm is consistent with
each other among WC2003, SC2006 and IDC. However, the absorption Ångström exponent from
SC2006 might be closer to the real ones than WC2003s as suggested in *Zhuang et al.* (*2015*). Therefore,
the SC2006 is adopted in this study.

The parameters in the correction procedure are derived from the local optical properties ($\omega 0$ $\omega_0$ and $\alpha s$

$\alpha_s$ were set to 0.922 and 1.51, respectively). The values of correction factors *C* and *R* are as follows: R=1

when ATN≤10 and *f*=1.2, and *C* in Nanjing is 2.95, 3.37, 3.56, 3.79, 3.99, 4.51 and 4.64 at 370, 470, 520,
590, 660, 880 and 950 nm. Detailed procedures of the calculations could be referred to *Zhuang et al.*
*(2015)*. Relatively in-depth description has been added in Section 2.2 in the revised manuscript.

**References:**
Collaud Coen, M., Weingartner, E., Apituley, A., Ceburnis, D., Fierz-Schmidhauser, R., Flentje, H.,
Henzing, J. S., Jennings, S. G., Moerman, M., Petzold, A., Schmid, O., Baltensperger, U.:
Minimizing light absorption measurement artifacts of the Aethalometer: evaluation of five correction
algorithms, Atmos. Meas. Tech. 2010, 3, 457–474.
Wu, D., Mao, J. T., Deng, X. J., Tie, X. X., Zhang, Y. H., Zeng, L. M., Li, F., Tan, H. B., Bi, X. Y.,
Huang, X. Y., Chen, J., Deng, T.: Black carbon aerosols and their radiative properties in the Pearl
River Delta region, Sci. China Ser. D, 52, 1152–1163, doi:10.1007/s11430-009-0115-y, 2009.
Wu, D., Wu, C., Liao, B., Chen, H., Wu, M., Li, F., Tan, H., Deng, T., Li, H., Jiang, D., Yu, J. Z.: Black
carbon over the South China Sea and in various continental locations in South China, Atmos. Chem.
Phys., 13, 12257–12270, doi:10.5194/acp-13-12257-2013, 2013.
Zhuang, B. L., Wang, T. J., Liu, J., Ma, Y., Yin, C. Q., Li, S., Xie, M., Han, Y., Zhu, J. L., Yang, X. Q.,
Fu, C. B.: Absorption coefficient of urban aerosol in Nanjing, west Yangtze River Delta, China, Atmos.
Chem. Phys., 15, 13633–13646, doi:10.5194/acp-15-13633-2015, 2015

4.  The truncation correction of Aurora-3000 is based on Mie calculations. If I understand it correctly, Mie calculation is not performed in this study, since obviously there is no measurement of particle number size distribution. Instead, correction parameters are directly taken from the literature in this study. How much uncertainty might be introduced to scattering coefficients due to the choice of the correction factors?

**R:** Thank you for your question. Mie calculation is not performed due to the lack of measurement of particle number size distribution at the site. However, in *Müller et al. (2011)*, it is pointed out that the calculation performed for Aurora-3000 in the study is accurate for a wide range of atmospheric aerosols, and the correction parameters have been used for correction in previous studies (e.g., *Virkkula et al., 2015*; *Gong et al., 2015*; *Perrone et al., 2014*; *Pandolfi et al., 2014*, etc.). For the single scattering albedo larger than 0.8, the uncertainty of the correction is not expected to be larger than 3 % (*Bond et al., 2009*). Here in our study, over 99% of the single scattering albedo is larger than 0.8, thus the uncertainty is about 2%~ 3%, which could meet the precision requirements to degrees.

R: Thanks for the comments.

Firstly, the existence of particulates could affect ozone photochemical production because particulates could inhibit the photolysis reactions near the surface in reducing the photolysis frequencies in the atmosphere, which would result in the decrease of $O_3$ concentrations near the ground. In our study, the negative correlation between particulates and $O_3$ coincides with the above assumption, which was also found in various numerical models (e.g., *Li et al., 2005*; *Bian et al., 2007*; *Deng et al., 2010*; *Li et al., 2011*; *Li et al., 2018, etc*.). Most of the simulated results above showed an obvious change in the amount of ozone concentration and production due to aerosols. For example, *Bian et al. (2007)* reported the ratio of $\Delta[O_3]/\Delta[AOD]$ ranged from -4~-16 ppb in Tianjin, and in *Li et al. (2011)*, aerosols decreased the average $O_3 \rightarrow O(^1D)$ photolysis frequency by 53%, 37% and 21% in the lower, middle and upper troposphere in central east China, and as implied in *Li et al. (2017)*, high concentrations of aerosols result in a 0.1~ 5.0 ppb (12.0%) reduction of near-surface ozone in central Nanjing.

Besides, we agree with you that the main reasons for the observed variation of ozone might be attributed to the effects of radiation, concentrations of precursors, other weather conditions, etc. And we have taken the effects above into consideration when discussing the ozone variation. For example, in Section 3.2, the discussion of ozone temporal variation contains the influence of radiation, precursor concentrations, as well as the meteorology field. And to make a better insight of the correlation and interaction between particles and ozone through observation data, this study further identifies the influence of associated affecting factors, including UV radiation, temperature, and precursors ($NO_x$, $NO_y$, and CO) concentrations, on the interaction (Section 3.3). For a more comprehensive overview, we not only analyze the correlation between particulates and ozone but also the one between particulates and the precursor ($NO_x$ and CO). It is found that particles ($PM_{2.5}$ and BC) are well-correlated with precursors ($NO_x$ and CO), which could be another possible reason for the negative correlation between aerosols and ozone. In our study, we have discussed the abovementioned possible reasons for the correlation thoroughly, instead of just laying emphasis on the impact of aerosols on the ozone photochemical production in the revised manuscript. Thus, the main points of our analysis and discussions is to propose the possible reasons for the effects of aerosols on ozone concentration (by influencing the radiation and the precursors concentrations) based on the observation data, rather than regard aerosols as a decisive factor of the observed ozone variation.

As for the analysis of data and deduction, according to your suggestion, Section 3.3 has been extended to degrees. More in-depth discussions on the aerosol classification and identification have been included in the current version. More details can be found in the revised manuscript.

with concentrated human activities with residential areas, schools, institutions and business districts, and
the main road of urban transportation around. Therefore, the results could suggest the characteristics and
interactions of pollutants in the urban region very well. Also, the results could further imply the effects
of the **urban underlying surface** and **human activities** to degrees. Besides, as a typical urban area, the
results in this study would probably bring new knowledge of aerosol characteristics, like the pollution
level variation in different years and different regions through comparison with previous studies based
on observation and numerical simulations.

Overall, this manuscript presents more comprehensive, systematic and deeper analysis on main pollutants
like particles and ozone in urban area of west YRD. Results further indicate the characteristics of the
particles and trace gases and reveal the possible chemistry process and interactions among different
species and meteorological variables in west YRD. And they are also advantageous to improve the
understanding of the detailed variations (seasonal, monthly, and diurnal) and its effects in east regions of
China.

According to your comments, questions and suggestions, not only the abstract, but the introduction, discussion and conclusion have also been rephrased carefully. The originality (listed above) and finding(s)
of this study have been refined in better ways of expression. Details can be found in the revised
manuscript.

The revised details could be referred to the new version of the manuscript, with relevant changes marked with traces.

[revised manuscript text omitted]
 come from the local region were from the places near Nanjing (cluster 2), north-west areasnorthern continent away from Nanjing (cluster 1), and northern regionsnorthern continent away far from Nanjing passing through oceans and urban agglomeration regions (cluster

4). Air masses from cluster 1 and cluster 2 both account for over 30% of the total aerosol characteristics and are more polluted with relatively high levels of particles, CO, and $NO_x$. Air masses in cluster 1 come from Shandong Province while those in cluster 2 come from local areas.

Particles and trace gases concentrations of cluster 2 are higher than those of cluster 1 to some extent, implying the severer air pollution problem in YRD region. This is different from that in SON.

Therefore, besides what has been discussed of cluster 3 and cluster 4 in SON, it is found that air

      The concentrations of O$_3$, similar to that in SON, is affected by radiation besides precursors levels. Thus, O$_3$ concentration in cluster 2 is a little higher than that in cluster 1.   The ratios of PM$_{2.5}$ to PM$_{10}$ of cluster 1 and cluster 2 are approximately equal in DJF, over 70%. The size of aerosols in cluster 1 and 2 are coarser, however, probably due to the higher RH (over 65%). Aerosols in cluster 1 are scatter to some extent compared to those in cluster 2. The trajectories of cluster 3 and cluster 4 are analogous to those in SON, respectively, but more polluted, probably due to more emissions in DJF especially in north China and weaker flow from ocean in DJF.

**3.5 Case Study**

For further understanding of the causes for high pollutants episodes, especially high particulate and O$_3$ episodes, detailed analysis of a typical episode from 2016 December 3-6 is presented in this section.

Fig.15 (a) and (b) show that high $O_3$ concentrations (over 80 ppb) occurred on December 4 with broad $O_3$ peaks (over 60 ppb) in the following days, while the average $O_3$ during the cold seasons was 37.7 ppb. Though there is a lack of particulate matter concentrations because of the instrument breakdown,  high concentrations of PMs might possibly occur referring to  the relatively high $\sigma_e$  value (over 500 Mm$^{-1}$) and BC concentrations (over 6

μg/m$^3$) on December 4th. Both PMs reach a maximum on December 5th (PM$_{2.5}$ over 200

μg/m$^3$ and PM$_{10}$ over 300 μg/m$^3$), over 3 times of the average concentrations. Besides, NO$_x$, NO$_y$, have reached high levels since December 4th (NO$_x$ over 70 ppb and NO$_y$ over 100 ppb). It is also noticeable that $\omega_0$  has a relatively sharp decrease from December 4th, especially on December

5th when particle concentrations were extremely high,  probably suggesting that the ratio of PM$_{10}$ became higher. Meanwhile, a relatively sharp increase occurred in $\alpha_{ts}$ , without any obvious variation in $\alpha_a$ , though, implying that scattering aerosols could take the leading role during this episode. It is also found that this case occurred under calm conditions before the passage of a cold front, which was  in the front of a continental high pressure system originating from Mongolia and sweeping over Nanjing (Fig.15 (c)).

And the decrease in temperature with high pressure system dominating eastern China  wes also detected on December 6th. Backward trajectory analysis for the past 96 hours (Fig.15

(d)) was conducted from December 5th at 20:00 LT, including the maximum concentrations of $O_3$ on December 4th and PMs on December 5th.  
[revised manuscript text omitted]